


# Comparison of four inverse modelling systems applied to the estimation of HFC-125, HFC-134a and SF$_6$ emissions over Europe

Dominik Brunner[1], Tim Arnold[2,3,4], Stephan Henne[1], Alistair Manning[2], Rona L. Thompson[5], Michela Maione[6], Simon O'Doherty[7], and Stefan Reimann[1]

[1]Laboratory for Air Pollution/Environmental Technology, Empa, Swiss Federal Laboratories for Materials Science and Technology, Dübendorf, 8600, Switzerland
[2]Met Office, Exeter EX1 3PB, United Kingdom
[3]National Physical Laboratory, Teddington, Middlesex TW11 0LW, UK
[4]School of GeoSciences, University of Edinburgh, Edinburgh, EH9 3FF, UK
[5]NILU - Norwegian Institute for Air Research, Kjeller, 2007, Norway
[6]Dipartimento di Scienze Pure e Applicate (DiSPeA), University of Urbino "Carlo Bo", Urbino, 61029, Italy
[7]School of Chemistry, University of Bristol, Bristol, UK

*Correspondence to*: Dominik Brunner (dominik.brunner@empa.ch)

**Abstract.** Hydrofluorocarbons (HFCs) are used in a range of industrial applications and have largely replaced previously used gases (CFCs and HCFCs). HFCs are not ozone depleting but have large global warming potentials and are, therefore, reported to the United Nations Framework Convention on Climate Change (UNFCCC). Here, we use four independent inverse models to estimate European emissions of the two HFCs contributing the most to global warming (HFC-134a and HFC-125) and of SF$_6$ for the year 2011. Using an ensemble of inverse models offers the possibility to better understand systematic uncertainties in inversions. All systems relied on the same measurement time series from Jungfraujoch (Switzerland), Mace Head (Ireland), and Monte Cimone (Italy), and the same a priori emissions, but differed in terms of Lagrangian transport model (FLEXPART, NAME), inversion method (Bayesian, Extended Kalman Filter), treatment of background mole fractions, spatial gridding, and a priori uncertainties. The model systems were compared with respect to the ability to reproduce the measurement time series, the spatial distribution of the posterior emissions, uncertainty reductions, and total emissions estimated for selected countries. All systems were able to reproduce the measurement time series very well with prior correlations between 0.5 and 0.9 and posterior correlations being higher by 0.05 to 0.1. For HFC-125, all models estimated higher emissions from Spain+Portugal than reported to UNFCCC (median higher by 390%). Estimates for Germany (+140%) and Ireland (+850%) were also considerably higher than UNFCCC, whereas the estimates for France and the UK were consistent with the national reports. In contrast to HFC-125, HFC-134a emissions from Spain+Portugal were broadly consistent with UNFCCC, and emissions from Germany were only 30% higher. The data suggests that the UK over reports its HFC-134a emissions to UNFCCC, as the model median emission was significantly lower, by 50%. An overestimation of both HFC-125 and HFC-134a emissions by about a factor 2 was also found for a group of eastern European countries (Czech Republic + Poland + Slovakia), though with less confidence since the measurement network has a low sensitivity to these countries. Consistent with UNFCCC, the models identified Germany as the highest national emitter of SF$_6$ in Europe, and the model median emission was only 1% lower than the UNFCCC numbers. In contrast, the model median emissions were 2-3 times higher than UNFCCC numbers for Italy, France and Spain+Portugal. The country-aggregated emissions from the different models often did not overlap within the range of the analytical uncertainties formally given by the inversion systems, suggesting that parametric and structural uncertainties are often dominant in the overall a posteriori uncertainty. The current European network of three routine monitoring sites for synthetic greenhouse gases has the potential to identify significant shortcomings in nationally reported emissions, but a denser network would be needed for more reliable monitoring of emissions of these important greenhouse gases across the whole of Europe.



## 1 Introduction

Synthetic halocarbons are used for a wide range of applications such as refrigeration and air conditioning, foams, solvents, aerosol products and fire protection. The first generation of compounds, the chlorine containing chlorofluorocarbons (CFCs) and bromine containing halons, were harmful to the stratospheric ozone layer and were phased-out under the Montreal

Protocol entering into force in 1987. They were substituted by natural refrigerants including hydrocarbons and ammonia and by another class of halocarbons, the hydro-chlorofluorocarbons (HCFCs), which have lower stratospheric ozone depletion potentials (ODPs) and lower global warming potentials (GWPs) than the CFCs. Regulation of the production and consumption of HCFCs under the Montreal Protocol led to a strong decline in their emissions over Europe after 2004 (Brunner et al., 2012; Derwent et al., 2007; Graziosi et al., 2015) whereas emissions were still increasing in developing

countries until recently (Saikawa et al., 2012; Xiang et al., 2014). Today, HCFCs and CFCs are mainly replaced by chlorine-free hydrofluorocarbons (HFCs), which are no longer harmful to the ozone layer except for minor indirect effects (Hurwitz et al., 2015), but some have large GWPs.

Current emissions of HFCs and CFCs are equivalent to only about 5% of global $CO_2$ emissions on a $CO_2$-equivalent basis, but, as Velders et al. (2009) highlighted, in a business as usual scenario without further regulations, HFC emissions could

grow to an equivalent of 9 – 19% of projected global $CO_2$ emissions by 2050, stressing the need for binding emission regulations. In view of the urgency of the problem and the success of the Paris Agreement, 197 countries adopted in October 2016 an amendment to the Montreal Protocol to phase down the emissions of HFCs by more than 80% over the next 30 years.

HFC-134a and HFC-125 considered in this study are the two most abundant HFCs in Europe constituting 69% of all HFC

emissions ($CO_2$-eq.) in 2012, with HFC-143a contributing another 23% according to officially reported emissions of the EU-28 countries. HFC-134a has a 100-yr GWP of 1,300 and is the preferred refrigerant in motor vehicle air conditioning systems. HFC-125 has a GWP of 3,170 and is mainly used in refrigerant blends for residential and commercial refrigeration and in smaller amounts as a fire suppression agent (O'Doherty et al., 2009; Velders et al., 2009). Sulfur hexafluoride ($SF_6$) is primarily used as a dielectric and insulator in high-voltage electronic installations. With a GWP of around 22,800, $SF_6$ is the

most potent greenhouse gas regulated reported to UNFCCC. $SF_6$ emissions are equivalent to about 0.5% of current global $CO_2$ emissions ($CO_2$-eq.), but emissions are still growing, especially in developing countries (Levin et al., 2010; Rigby et al., 2010).

Due to their long atmospheric lifetime, HFCs and $SF_6$ are rather uniformly distributed in the atmosphere. Global emissions can, therefore, be estimated from measurements at a few representative baseline stations distributed across the globe

(Cunnold et al., 1994; Montzka et al., 2015; Vollmer et al., 2011; Xiang et al., 2014). Estimating emissions at continental or even regional and country scale, however, requires a denser network of sites with varying sensitivity to emissions from the region of interest (Villani et al., 2010).

Currently, HFCs are routinely measured at only three sites in Europe: Jungfraujoch in Switzerland, Mace Head in Ireland, and Monte Cimone in Italy. Measurements from these sites have been used in several previous inverse modeling studies to

estimate European emissions of selected halocarbons and $SF_6$ (Brunner et al., 2012; Ganesan et al., 2014; Keller et al., 2011; Keller et al., 2012; Lunt et al., 2015; Maione et al., 2014; Manning, 2011; Manning et al., 2003; Rigby et al., 2011; Simmonds et al., 2016; Stohl et al., 2009). Different Lagrangian transport models and inversion approaches have been applied in these studies but no systematic comparison between the model systems has been undertaken so far. The European infrastructure project InGOS (Integrated non-$CO_2$ Greenhouse gas Observation System) helped to improve the quality and

compatibility of these measurements, to further develop the measurement technologies, and to collect and harmonize the




data. It also supported a range of modeling studies to quantify European emissions of non-$CO_2$ greenhouse gases including $CH_4$ and $N_2O$ (Bergamaschi et al., 2015) and halocarbons (this study), and to evaluate the models with respect to their transport properties.

Inverse emission estimation using direct atmospheric observations (commonly referred to as 'top-down') has been proposed as a tool for helping to verify anthropogenic emission inventories estimated by the individual countries based on statistical data and source-specific emission factors (commonly referred to as 'bottom-up') (Nisbet and Weiss, 2010). However, to enhance the credibility of this top-down approach, a better understanding of the associated uncertainties is needed. Currently, there is no commonly accepted benchmark against which to test the models and there is no single emission source that is known well enough to serve this purpose. Emissions of radon, for example, have turned out to be spatially and temporally more variable than previously thought (Karstens et al., 2015). Large-scale tracer release experiments such as ETEX (Van dop et al., 1998) have been instrumental in the development of dispersion models, but their temporal and spatial coverage is too sparse for an overall assessment of atmospheric transport and inverse modeling systems. Traditionally, inverse modeling studies have applied a single transport model and inversion setup and reported posterior uncertainties deduced from Gaussian error statistics in a Bayesian framework. More recently, awareness has grown that this approach may miss important contributions to the true uncertainties, including errors in model transport, representation errors, and uncertainties related to the chosen setup and the expert judgments that classical Bayesian inversions heavily rely on. Approaches to overcome these limitations included a better consideration of transport uncertainties (Baker et al., 2006; Lin and Gerbig, 2005; Locatelli et al., 2013), objective estimation of error covariance parameters (Berchet et al., 2013; Brunner et al., 2012; Michalak et al., 2005), and model experiments exploring the sensitivity of the results to different assumptions (Bergamaschi et al., 2010; Brunner et al., 2012; Henne et al., 2016). A promising new avenue is to extend the classical Bayesian framework with the dimension of 'uncertainties of uncertainties' (Berchet et al., 2015; Ganesan et al., 2014).

Here we apply four independent inversion systems to quantify the emissions of HFC-134a, HFC-125 and $SF_6$ over Europe for the year 2011 in a set of well-defined model experiments with common observation data and a priori emissions. We aim to compare the results of four well-established systems used in previous studies and to better assess the uncertainties associated with different choices of transport model, inversion method, treatment of background mole fractions, spatial gridding, a priori uncertainties, and error correlation structures, which add to the analytical uncertainties determined by the individual systems. Furthermore, we aim to evaluate the ability of the current network of three monitoring sites in Europe to constrain the emissions of synthetic greenhouse gases in individual European countries.

## 2 Methods

### 2.1 Observation Data

Measurements were available as hourly or two-hourly samples from the coastal site, Mace Head (9.90°W, 53.33°N, 15 m above mean sea level (amsl)), Ireland, and the two mountain sites, Jungfraujoch (7.99°E, 46.55°N, 3573 m amsl), Switzerland, and Monte Cimone (10.70°E, 44.18°N, 2165 m amsl), Italy. Halocarbons and $SF_6$ are measured at Jungfraujoch and Mace Head with a 'Medusa' Gas Chromatography/Mass Spectrometry (GC/MS) system (Miller et al., 2008). At Monte Cimone, an Adsorption Desorption System (ADS) GC/MS (Maione et al., 2013) is used, which does not enable $SF_6$ to be measured. The measurement data and their uncertainties (1σ single measurement precision determined as running mean of calibration standards bracketing each measurement) were provided to all groups at their native time resolution. Typical precisions for HFC-134a, HFC-125 and $SF_6$ are in the range 0.2-0.5 ppt, 0.05-0.1 ppt and 0.02-0.03 ppt, respectively.




For the assimilation, these observations were averaged to 3-hourly values in the EMPA and EMPA2 models and to daily means in NILU. UKMO used a single 3-hourly mean value per day around the time when the uncertainty of boundary layer heights was considered to be lowest, i.e. in the early afternoon (12-15 UTC) at Mace Head, and when the least influence from local boundary layer transport can be expected at the two mountain sites (06-09 UTC).

**2.2 Inverse Modelling Systems**

A brief overview of the four inversion systems employed in this study is presented in Table 1. All systems have been used in similar configurations in previous studies as referenced in the table. In all systems, atmospheric transport was described by a Lagrangian Particle Dispersion Model (LPDM). The LPDMs were operated in backwards in time, receptor-oriented mode (Seibert and Frank, 2004). In this mode, virtual particles (infinitesimally small air parcels) are released at the measurement sites and followed backwards in time, typically for a few days.

Three systems (EMPA, EMPA2, NILU) used the transport model FLEXPART (Stohl et al., 2005) driven by 3-hourly analysis and forecast fields from the European Centre for Medium Range Weather Forecasts - Integrated Forecast System (ECMWF-IFS). The fourth system, UKMO, relied on the transport model NAME (Ryall and Maryon, 1998) driven by global analyses of the UK Met Office's Numerical Weather Prediction model.

The outputs of the LPDMs are emission sensitivity maps, so-called 'footprints', for each particle ensemble release time. The footprints represent the total sensitivity of an observation to surface emissions over the backwards simulation time. Multiplying the footprint by an emission map and integrating in space and time gives a simulated mole fraction at each release time and location. Assuming temporally constant emissions for the inversion period, the relation between emissions and simulated mole fractions can be written as

$$\mathbf{y} = \mathbf{M}\mathbf{x}, \tag{1}$$

where $\mathbf{y} = (y_1 \quad \cdots \quad y_m)$ is the vector of simulated mole fractions at all times and stations, with $m$ being the total number of available measurements. $\mathbf{x} = (x_1 \quad \cdots \quad x_n)$ is the state vector which includes the gridded emissions and possibly other elements such as background mole fractions (see below), and $n$ is the number of state vector elements to be estimated/optimized by the inversion. $\mathbf{M}$ is the sensitivity matrix (with dimension $m$ x $n$),

$$\mathbf{M} = \begin{pmatrix} M_{1,1} & \cdots & M_{1,n} \\ \vdots & \ddots & \vdots \\ M_{m,1} & \cdots & M_{m,n} \end{pmatrix}. \tag{2}$$

Each row of $\mathbf{M}$ describes the sensitivity of a given measurement to all state vector elements composed of the footprint computed by the LPDM and possibly other elements such as the sensitivity to the background field (see e.g. Thompson and Stohl, 2014).

The goal of the inversion is to estimate an optimized state $\mathbf{x}$, which accounts for the observed mole fractions $\mathbf{y}_o$ by reducing the difference between observed and simulated values, additionally constrained by the uncertainty bounds of the prior state variables. In the Bayesian framework and assuming Gaussian uncertainty distributions, this optimized state is obtained by minimizing the following cost function $J(\mathbf{x})$ (e.g. Tarantola, 2005)

$$J(\mathbf{x}) = \frac{1}{2}(\mathbf{x} - \mathbf{x}_b)^{\mathrm{T}}\mathbf{B}^{-1}(\mathbf{x} - \mathbf{x}_b) + \frac{1}{2}(\mathbf{M}\mathbf{x} - \mathbf{y}_o)^{\mathrm{T}}\mathbf{R}^{-1}(\mathbf{M}\mathbf{x} - \mathbf{y}_o). \tag{3}$$




The first term on the right-hand side describes the deviation of the optimized state **x** from a prior state **x**$_b$, the second term the deviation of the simulated mole fractions from the observations. Both terms are weighted by their uncertainties represented by the error covariance matrices **B** ($n$ x $n$) and **R** ($m$ x $m$) for the prior and observation uncertainties, respectively.

This approach was employed by the inversion systems EMPA2, NILU and UKMO, which, however, differed in various

other aspects of the implementation. In order to mimic the approach presented by Stohl et al. (2009) as closely as possible, EMPA2 assumed the matrices **B** and **R** to be diagonal (i.e., uncorrelated errors). NILU, instead, assumed a correlation length scale of 200 km over land and 1000 km over ocean for the prior emission field, and **R** contained off-diagonal elements to represent the cross-correlations of the model representation error (see Thompson and Stohl, 2014). Like EMPA2, UKMO did not account for potentially correlated errors in the prior emission field. As will be shown in Sect. 3, the choice of correlation

structure has quite a strong influence on the results. Due to the way bottom-up inventories are generated, it may be justified to assume stronger error correlations within a country than across country borders, but none of the inversion systems adopted such a strategy.

To avoid non-physical negative emissions, NILU applied a 'truncated Gaussian' approach (Thacker, 2007; Thompson and Stohl, 2014). This entails performing a second step after the inversion in which an inequality constraint, namely that the

emissions must be greater than or equal to zero, is applied accounting also for the error-covariance between grid-cells.

EMPA2 estimated the model uncertainty following the suggestions by Stohl et al. (2009). In a first step, the Root Mean Square Error (RMSE) of the prior simulation minus observations was calculated for each site separately. The model residuals were then scaled by the RMSE. The normalized residual distribution often does not follow a normal distribution, but is skewed towards large negative values (large model underestimations). In order to reduce the influence of such points in the

inversion, the model uncertainty for these 'outliers' was iteratively adjusted so that the normalized residual distribution followed a normal distribution more closely. This procedure was repeated using the posterior simulations of a first inversion run. A second and third inversion run was then performed using the updated model uncertainties but the same prior state. Furthermore, prior uncertainties were reduced for grid cells with negative posterior emissions, and the inversion was iterated until a solution without significant negative emission contributions was obtained, again following the suggestion by Stohl et

al. (2009).

The Met Office's inverse modelling system (InTEM – Inversion Technique for Emission Modelling) using the NAME model has evolved since the work of Manning et al. (2011) and the NitroEurope project (Bergamaschi et al., 2015) and is now based on a Bayesian methodology. Measurement uncertainty reported in the InGOS data set was used as observation error. Model-measurement mismatch errors were also applied to each measurement and were calculated using a metric based on

the degree of influence of local fluxes on the measurement (Manning et al., 2011). These model errors were inflated based on the difference between the model release height above sea level and the true altitude of the observation, and the relative difference between the modelled boundary layer height and the observation height. No spatial or temporal correlations were applied in these inversions. Grid boxes were aggregated based on the sensitivity of measurements to emissions, creating around 100-150 course grid regions within the inversion domain. A non-negative least square solver was used to optimise

the solution thus preventing negative emissions from being estimated.

EMPA applied an extended Kalman Filter (ExtKF) as described in detail in Brunner et al. (2012). Different from the other systems the observations are not used all at the same time, but are assimilated sequentially thereby gradually adjusting the state to a solution that is optimal given all past observations up to the assimilation time. The Kalman filter update equations are for the state:





$$\mathbf{x}_k^+ = \mathbf{x}_k^- + \mathbf{K}_k(\mathbf{y}_k - \mathbf{M}_k\mathbf{x}_k^-) \tag{4}$$

and for the uncertainty of the state:

$$\mathbf{P}_k^+ = (\mathbf{1} - \mathbf{K}_k\mathbf{M}_k)\mathbf{P}_k^- \tag{5}$$

where $k$ is the time index, $\mathbf{K}_k$ the Kalman Gain matrix, defined as

$$\mathbf{K}_k = \mathbf{P}_k^-\mathbf{M}_k^{\mathrm{T}}\left(\mathbf{R}_k - \mathbf{M}_k\mathbf{P}_k^-\mathbf{M}_k^{\mathrm{T}}\right)^{-1}, \tag{6}$$

$\mathbf{P}_k$ the state error covariance matrix, and $\mathbf{M}_k$ the sensitivity matrix for time $k$. The minus sign denotes a 'first guess' state before assimilation of the observation $\mathbf{y}_k$ available at time $k$, and the plus sign denotes the 'analysis' state after assimilation.

The matrix $\mathbf{P}$ essentially takes the role of $\mathbf{B}$ in the Bayesian inversion and the observation and model representation uncertainty matrix $\mathbf{R}$ is included in the definition of the Kalman Gain matrix. The similarity between the Kalman Filter and Bayesian inversion is further illustrated by the fact that the solution to Eq. (3) is given by the same Eq. (4) but with $\mathbf{B}$ replacing $\mathbf{P}_k^-$ in the Kalman Gain matrix and all observations being used at once instead of looping over time steps $k$. Different from the Bayesian inversions, however, the emissions were not assumed to be constant but to evolve slowly with

time as expressed by the forecast equation

$$\mathbf{x}_{k+1}^- = \mathbf{x}_k^+ + \boldsymbol{\varepsilon}_k, \tag{7}$$

which states that the emissions at time $k+1$ are expected to be the same as at time $k$ within an uncertainty $\varepsilon_k$. This step adds uncertainty to the emissions according to

$$\mathbf{P}_{k+1}^- = \mathbf{P}_k^+ + \mathbf{Q}_k, \tag{8}$$

so that the uncertainty can grow with time in regions poorly covered by the observations. This is different from the other inversions where the posterior uncertainties are always smaller than the prior uncertainties. Without this forecast step, the

solution after assimilating all observations would be identical to the solution obtained with Eq. (3). The new matrix $\mathbf{Q}_k$, which has no correspondence in the Bayesian inversion, describes the uncertainty of the forecast and determines how rapidly the emissions (and background levels, see below) are allowed to change with time.

Another unique feature of the EMPA system is that it estimates the logarithm of the emissions in order to constrain the solution to positive values. This makes the problem non-linear and, therefore, requires the application of an Extended

Kalman Filter that linearizes the sensitivity matrix around the current state. An important effect of this approach is that the residuals $(\mathbf{y}_k - \mathbf{M}_k\mathbf{x}_k^-)$ become approximately normally distributed, a prerequisite for the Kalman Filter to provide an optimal solution. Finally, temporal correlations in the residuals were accounted for by applying an augmented state red-noise Kalman Filter as described in Brunner et al. (2012).

### 2.3 Background Treatment

The mole fractions of an inert trace gas at any given point in the atmosphere may be considered to be composed of a smoothly varying, large-scale background plus a more rapidly varying component containing the imprint of recent sources and sinks. Since the LPDM simulations only account for the contribution from recent emissions (the time period covered by





the backward simulations), the background has to be treated separately. All inversion systems estimated a prior background, and three of the four systems optimized the background along with the emissions, but the details of this optimization differed.

For the prior background mole fractions, NILU used the method described in Thompson and Stohl (2014). In brief, this involved the following three steps: 1) selecting observations defined to be representative of the background, i.e., lower quartile of values in a shifting time window of 60 days (30 days for $SF_6$), 2) calculating the contribution to these observations from prior emissions within the domain and subtracting these, and 3) interpolating the background mole fractions to the observation time step.

EMPA2 applied the Robust Estimation of Baseline Signal (REBS) method (Ruckstuhl et al., 2012), which iteratively fits a non-parametric local regression curve to the observations, successively excluding points outside a certain range around the baseline curve. REBS was applied separately to individual observations from each site using asymmetric robustness weights with a tuning factor of $b = 2.5$, a temporal window width of 60 days and a maximum of 10 iterations. An estimate of the baseline uncertainty is given by REBS as a constant value for the whole time series.

In the UKMO set up, a total of eleven extra 'boundary condition' variables were estimated as part of the inversion. The prior background time-series was calculated using data at Mace Head when well-mixed 'clean' air arrived from the North Atlantic Ocean. The eleven variables are multiplication factors to calculate the mole fractions of the background air arriving from eight horizontal (SSE, SSW, WSW,…, ESE) boundaries at 0-6 km, two boundaries (north and south) from 6 to 9 km, and a boundary at 9 km (upper troposphere to stratosphere).

EMPA2 optimized the background levels separately for each measurement site at selected reference points every 14 days. Background levels in between these reference points were linearly interpolated. NILU did not optimize the background to avoid cross-talk between the optimization of the emissions and the baseline. In the EMPA system, a single element per observation site is added to the state vector to represent the background at time step $k$. This background is then allowed to evolve slowly with time similar to the evolution of the emissions. As first guess for the initialization of the assimilation, the $5^{th}$ percentile of the first 12 days of measurements is used.

## 2.4 Inversion Grids

In order to limit the dimension of the problem, all four systems feature a reduced resolution grid to represent the emissions in the state vector. EMPA and EMPA2 computed a reduced grid by iteratively aggregating grid cells until the enlarged cell passed a threshold with respect to its annual mean total surface sensitivity. The result of this procedure is illustrated in Figure 1, which also presents the position of the three measurement sites and the common domain chosen for the inversion.

NILU employed a reduced grid based on the emission sensitivity with a maximum resolution of 1°x1° over land (effectively most of Europe is resolved at 1°x1° and larger grid cells are only found in Eastern Europe), and a resolution of 4°x4° over sea. UKMO used a grid that follows the outlines of countries or groups of countries of interest, which ensures that parts of different countries are prevented from being aggregated into the same coarse grid. Within country, grid cells can be split further depending on the sensitivity of the measurements to emissions from such areas.

## 2.5 Experiments

All experiments and required outputs were described in a detailed modelling protocol available to the participants. Three main experiments (M1-M3) were defined to estimate the emissions of HFC-125, HFC-134a, and $SF_6$, respectively. For HFC-125, several additional experiments were defined to test the sensitivity to changing prior uncertainty, background treatment, data selection, and uniform versus spatially resolved prior emissions. Most of these sensitivity tests were limited to a single





inversion system. A summary of the main and sensitivity experiments is presented in Table 2. All experiments were performed for a single year (2011) and the main scope was the estimation of annual mean emissions.

To make the results as comparable as possible, all inversion systems used the same observation data (including uncertainties) and prior emissions, and the backward transport simulations were started from the same horizontal coordinates. Since the comparatively coarse topography in the transport models significantly underestimates the true altitude of the two mountains sites, particles were released at 3000 m amsl at Jungfraujoch and at 2000 m amsl at Monte Cimone, thus a few hundred meters below the true station height but still well above the model topography. Previous analyses of FLEXPART simulations indicated that 3000 m amsl is an optimal release height for Jungfraujoch at the given model resolution of 0.2° x 0.2° (Brunner et al., 2012). However, for the NAME model it turned out that a release height of 3000 m amsl. overestimates the sensitivity to regions surrounding Jungfraujoch, especially France. For NAME a significantly higher release height of 2000 m above model ground (which corresponds to 3906 m amsl) was selected to provide footprint sensitivities comparable to those of FLEXPART.

In order to preserve the characteristics of the individual inversion systems as used in previous studies, no further common settings were specified. In particular, the groups were free to choose the inversion grid, the prior uncertainties (except for experiment FLAT) and error correlation structures (see Table 2). Model outputs defined by the protocol included simulated time series at the measurement sites, gridded emission fields, and estimates of country-aggregated emissions. These outputs form the basis of the results presented in the following.

## 3. Results and Discussion

### 3.1 Simulated Time Series

Simulated prior and posterior time series at all three measurement sites are shown in Figure 2 and 3 for HFC-125 mole fractions for experiment M1 (definition see Table 2). Corresponding figures for M2 (HFC-134a) and M3 ($SF_6$) are presented in the supplementary material.

The simulations successfully reproduce much of the observed variability, indicating that the underlying variations in meteorology and atmospheric transport are well represented by the models. The variance explained by the prior time series ranges between 30% and 80% depending on the site (lowest at Monte Cimone, highest at Mace Head) and the LPDM, and is further increased in the posterior time series. The alternation between clean Atlantic air and advection of polluted air masses from UK and the European continent observed at Mace Head is very well matched by all models. The largest difference between the models is the representation of background concentrations, with NILU being lower than the other models towards the end of the one-year period at Mace Head. The two mountain sites Jungfraujoch and Monte Cimone are more frequently perturbed by polluted air masses and the background level is less clearly defined. As a consequence, the scatter between the background levels is rather large with UKMO tending to be at the lower and EMPA at the upper end of the estimates. Note, however, that EMPA does not have a prior background in the same way as the other models since its background is constructed directly during the assimilation process. The prior mole fractions shown in Figure 2, therefore, have been added to the posterior background in the case of EMPA.

Although many of the peaks observed at the two mountain sites are well captured, reproducing the observations is more challenging at these sites compared to Mace Head. At all three sites, the performance of the posterior simulations is clearly improved and the spread between model simulated peaks and background levels is reduced.

The overall model performances in experiments M1-M3 are summarized in Figure 4 in the form of Taylor diagrams. For HFC-125, the diagrams confirm the qualitative picture presented above: Mace Head is simulated best with posterior





correlations between 0.8 and 0.92, compared to values in the range of 0.6 to 0.82 at the mountain sites. The posterior scores are closer to each other than the prior scores. In particular, the score of NAME is moving closer to the three FLEXPART-based systems. For HFC-134a, the posterior performances are similar as for HFC-125 except for Monte Cimone where all models have difficulties in reproducing the observations. While the prior simulations of HFC-125 showed too little variance

at Jungfraujoch and Mace Head suggesting that emissions in the surroundings of these sites were underestimated, the prior simulations of HFC-134a tended to be too high. Observations of $SF_6$ were only available from Jungfraujoch and Mace Head. $SF_6$ is very well simulated at these sites such that the improvement from prior to posterior is relatively small.

Overall, the FLEXPART-based systems performed somewhat better than the UKMO system. This is especially true for Jungfraujoch whereas at Mace Head the differences were minor. The reasons for this are unclear: Differences in the

dispersion model, the underlying meteorological model, and/or model setup (e.g. particle release height) are all potential candidates for further study.

**3.2 Gridded emissions**

Gridded prior emissions are exemplarily presented in Figure 5 for HFC-134a (experiment M2). Although based on exactly the same EDGAR v4.2 inventory data, the spatial aggregation to the different inversion grids leads to visually quite different

distributions. The UKMO grid, for example, is rather coarse and follows the country outlines. The grids of NAME, EMPA and EMPA2 have higher resolution near the observation sites and lower resolution further away. NILU has a nearly constant resolution over land and reduced resolution over the sea. These different grids combined with different a priori uncertainties and correlation length scales will influence the inversion results as they offer different flexibility to optimize the emissions. Further insights into these sensitivities will be presented in Sect. 3.4 (country aggregated emissions).

The emission updates, i.e., the posterior minus prior emissions are shown in Figures 6–8 for experiments M1 to M3. For HFC-125, the posterior differences share a number of similarities between the models such as positive values over the Iberian Peninsula, mid and southern Italy, western France, south-western UK, and negative values over northern Italy and northern/north-eastern UK. Overall, EMPA and EMPA2 are quite similar except for opposing patterns over the Benelux countries and south-eastern UK. NILU estimates much larger enhancements over Spain than the other models. It also finds

significant enhancements in a band extending from Germany towards the Baltic countries, where the other models find either small (UKMO) or even negative increments (EMPA, EMPA2). These rather large differences are somewhat surprising considering the fact that the posterior time series simulated by the models are of similar quality (Figure 3). A notable difference between the models is the consistently lower background in the NILU system at Mace Head between October and December, probably because it does not optimize the background in the inversion. However, the sensitivity test NOBLOPT

(Table 2, results in Sect. 3.4), where EMPA2 repeated the experiment without background adjustment, still showed large differences from NILU in this period, suggesting that they were already present in the prior background. In the case of no background optimisation, emissions estimated by EMPA2 were generally higher in most of the domain (total of 1.1 Gg/yr higher) as compared with the reference run M1. Differences were especially large for the Iberian Peninsula and Italy, but not towards north-eastern Europe as in NILU.

A similar picture emerges for HFC-134a (Figure 7). The models estimate reductions with respect to the prior emissions over eastern and northern UK and northern Italy. All models find enhanced posterior emissions over Spain and Portugal with NILU estimating again the largest changes, similar to HFC-125. For Germany, there is little consistency between the models. While NILU and EMPA show reductions over the western and increases over the eastern parts of the country, EMPA2 estimates a uniform reduction and UKMO finds decreases in the northern and increases in the southern parts. A unique

feature of NILU is again a band of positive changes extending from Germany to the Baltic countries. UKMO simulates a




pronounced dipole pattern in the area of Paris. Such dipole patterns occur more easily when spatial correlations in the prior uncertainties are not considered.

For $SF_6$, all models consistently simulate lower posterior than prior emissions over Germany, the country with the largest emissions of $SF_6$ in Europe. Except for UKMO, the models consistently find increased emissions in Italy and the western

parts of France. Similar to HFC-125 and HFC-134a but different from the other systems, NILU simulates strong enhancements for the Iberian Peninsula. Most models find a local reduction around Jungfraujoch, especially UKMO.

### 3.3 Uncertainty reductions

A useful diagnostic of the model results is the uncertainty reduction as it illustrates the influence of the measurements on the posterior fields. Furthermore, the uncertainty reduction depends on the magnitude and correlation structure of the prior

uncertainties and thus helps illustrating the effect of the different model choices.

Figure 9 presents the absolute prior uncertainties chosen in the four systems for the example of HFC-134a. Corresponding figures for HFC-125 and $SF_6$ are provided in the supplement. EMPA and EMPA2 specified the uncertainties relative to the prior emissions. As a result, the distribution closely follows the pattern of prior emissions. This is also true for UKMO although uncertainties in grid cells with very low emissions were set to a minimum value. Overall, much lower prior

uncertainties were specified in EMPA and EMPA2 compared to NILU and UKMO. In EMPA, the relative uncertainties were set to a range of about 70% for the largest and 100% for the smallest grid cells, accounting for the assumed uncertainty correlation length of 500 km. In EMPA2, the uncertainties were set uniformly to 137%, but to prevent negative emissions, these uncertainties had to be reduced iteratively in some grid cells. UKMO assumed a 200% uncertainty in the prior emissions plus a minimum value. In NILU the uncertainties for each grid cell were set to 100% of the largest emission out of

itself and the 8 neighbouring grid cells and in addition, a minimum uncertainty was specified. This was done to allow a higher degree of freedom in adjusting the spatial pattern of emissions.

Together with the different spatial uncertainty correlations, these differences have a marked effect on the resulting uncertainty reductions. Figure 10 shows the reductions achieved for HFC-134a. Uncertainty reductions are largest and rather uniform for NILU due to the large prior uncertainties and prior error correlations with a length scale of 200 km over land.

Almost no reductions are found over sea due to very low prior uncertainties. Uncertainty reductions are more scattered in EMPA2 due to the absence of spatial correlations in the prior error covariance matrix. The pattern reflects a combination of the influence of the measurements and magnitude of the prior fluxes. Largest reductions tend to occur in grid cells with large prior emissions. Due to the growing cell sizes with increasing distance from the measurements, error reductions do not fall off as clearly with distance from the sites as in the NILU system.

Uncertainty reductions are only moderate in UKMO despite rather large prior uncertainties. This is likely due to an eight times smaller number of observations assimilated (one morning or afternoon value instead of eight 3-hourly values per day) compared to EMPA and EMPA2 and larger assumed data-mismatch uncertainties, especially compared to NILU. The data-mismatch uncertainties adopted for Mace Head, for example, correspond to average HFC-134a mole fraction uncertainties of 1.9 ppt for EMPA and EMPA2, 1.2 ppt for NILU, and 3.4 ppt for UKMO, respectively. At Jungfraujoch, the uncertainty

specified in UKMO was about 5 times larger than in the other models, reflecting the high uncertainty in simulated transport assumed for this site. Note that in all inversion systems the data-mismatch uncertainty is much larger than the stated measurement precision and is thus dominated by representation and transport model uncertainties.

Due to the optimization of the logarithm of emissions, the EMPA system reduces relative rather than absolute uncertainties. The uncertainty reduction is, therefore, presented in terms of reduction of relative uncertainties. The uncertainty reductions

are typically between 40% and 70%. Similar to EMPA2, the uncertainty reductions do not fall off strongly with distance



from the sites due to the irregular grid. Unlike EMPA2, however, the pattern is much more uniform due to the consideration of spatial error correlations. Minor maxima coincide with grid cells with large prior emissions.

### 3.4 Country aggregated emissions

An important question in the context of international treaties such as the recent Paris Agreement is the question, how suitable
is the current observation network to constrain emissions at the country level? For this purpose, the gridded emission fields were aggregated to individual countries or groups of countries. Due to the relatively coarse grids, this aggregation can be a significant source of error. Emissions from grid cells covering two or more countries need to be properly assigned to the individual countries. This was done either by weighting according to the fractional area covered by each country (EMPA, NILU), or by weighting according to the relative share of the population in the overlapping cell using high-resolution
population density data (EMPA2). UKMO circumvented the problem by specifying a grid following the country borders.
Another critical question is whether emissions from grid cells covering both land and sea should be fully assigned to the land areas or whether only the fraction covered by land should be considered. This is particularly relevant for countries like Italy with long coastlines and for inversion grids with large cells. In all models it was assumed that emissions from grid cells partially overlapping sea areas are fully assigned to the adjacent land areas assuming that emissions over sea are negligible.
UKMO explicitly extended the country masks to include offshore sea areas.
Figure 11 presents the prior emissions of HFC-125 estimated by the four model systems. Differences between these estimates reflect the uncertainty introduced by the different grids and country attribution strategies. These differences are typically in the range of 1% to 6% of the country emissions but occasionally can be larger. For Denmark, for example, the values vary between a minimum of 32 Mg/yr (EMPA) and 120 Mg/yr (UKMO). The low value estimated by EMPA is
largely attributable to the area of Copenhagen being part of a large grid cell also covering large parts of southern Sweden resulting in a significant misattribution of emissions from Denmark to Sweden. As a consequence, emissions from SW+FI+BALT are relatively high in this model. Estimates of EMPA2 and UKMO are generally very close to each other suggesting that the usage of high-resolution population density data for redistributing sub-grid cell emissions is nearly equivalent to using a grid following the country outlines.
The corresponding posterior estimates for HFC-125 are shown in Figure 12. Here, the differences between the models are much larger. EMPA and NILU have larger adjustments with respect to the prior than the other two models; integrated over all countries their emissions are about 50% higher. The standard deviation between the four model estimates for the domain total is 26%. NILU estimates particularly large enhancements for Germany, the Iberian countries ES+PT, the Nordic countries SW+FI+BALT, and the eastern European countries PO+CZ+SV, consistent with the spatial pattern in Figure 6.
EMPA, conversely, estimates only small changes for Germany, similarly large enhancements for ES+PO, and uniquely large enhancements for Italy and the Benelux countries BE+NL+LU. The stronger adjustments in EMPA and NILU are likely related to the spatial error correlations considered in these systems but also to other factors (see Sect. 3.5).
Rather than considering the models individually, they may also be treated as an ensemble of estimates that can be compared to the bottom-up emissions officially reported to UNFCCC. A summary of this comparison for the experiments M1-M3 as
well as the sensitivity experiment FLAT (discussed in Sect. 3.5) is presented in Figure 13. Shown are median values for the prior and posterior model estimates as well as the range between minimum and maximum. For HFC-125 (panel a) there is a rather high consistency between the top-down estimates and the UNFCCC values for many countries including FR, IT, UK, and Benelux. Marked differences with all models being either higher or lower than UNFCCC are found for DE (model median is 2.4x higher than UNFCCC), ES+PT (4.9x higher), IR (9.5x higher), SW+FI+BALT (2x higher), PO+CZ+SV
(2.8x smaller), and CH (2x smaller). It should be noted that the prior emissions based on the EDGAR v4.2 2008 inventory



for HFC-125 are significantly different from the UNFCCC 2011 emissions officially reported by the countries (grey bars). This is especially true for the countries DE and PO+CZ+SV, where the posterior model estimates are closer to the EDGAR prior. The estimated significant underestimation of the HFC-125 emissions reported to UNFCCC by Ireland and Spain+Portugal, that was consistently found across all model systems, has also been reported previously by Brunner et al.

(2012). Summed over all countries, the model median estimate is 24% higher than the UNFCCC total. For some countries, our results can also be compared with those by Lunt et al. (2015), which covered a similar period (2010-2012) and also used EDGAR as prior (see their Table S3). For example, they also found higher than UNFCCC emissions from Germany though not as large as EDGAR. For France their posterior remained close to EDGAR and was lower than UNFCCC. Emissions from UK and Italy were significantly increased which is in contrast to our results.

For HFC-134a, the model estimates are generally more consistent with UNFCCC than for HFC-125 (Figure 13c). In strong contrast to HFC-125, this is also true for Ireland and Spain+Portugal. The high consistency also applies to the domain total, which is only 11% lower than the total reported to UNFCCC. For SW+FI+BALT and PO+CZ+SV there are similar discrepancies as for HFC-125. Again, this is at least partly caused by the large differences between the prior and UNFCCC emissions and the large influence of the prior on the final model estimates. The model estimates are consistently lower than

the UNFCCC values for UK by about a factor of two, which contributes strongly to the 11% difference for the domain total. An overestimation of the HFC-134a emissions reported by UK has also been found previously by Lunt et al. (2015) and Say et al. (2016) and is in part due to the use of an assumed high loss rate of HFC-134a from car air conditioning systems in the UK. For Italy, the model estimates are consistently higher than the UNFCCC values by 40% on average. Note, however, that the results for Italy are strongly influenced by the measurements at Monte Cimone where the models had difficulties in

reproducing the HFC-134a measurements. Lunt et al. (2015) found an even stronger increase over Italy (factor 2.4), whereas they obtained relatively consistent (compared with UNFCCC) estimates for Germany and reductions by ~25 % in France, in fair agreement with our results.

For emissions of $SF_6$ the attribution to the different countries is very different from HFC-125 and HFC-134 (Fig. 13d). Consistent with the bottom-up estimates reported to UNFCCC, the models identify Germany as the highest national emitter

in Europe. The model median is highly consistent with UNFCCC but almost a factor 2 lower than the EDGAR v4.2 prior. For almost all other countries, however, the model estimates are closer to EDGAR v4.2 than to UNFCCC. For Italy, France, and Spain+Portugal, for example, the model medians are a factor 2-3 higher than the UNFCCC values but very close to EDGAR v4.2. Summed over all countries, the models are 47% higher than UNFCCC. $SF_6$ emissions have also been estimated by Ganesan et al. (2014) for the year 2012 based on a slightly modified EDGAR4.2 prior. Their estimates for

Germany (348 Mg/yr) were much higher than ours (137 Mg/yr), but also their prior was much higher (650 Mg/yr compared to 254 Mg/yr). We note that our prior (obtained as a sum over all grid cells covering Germany) is consistent with the country table provided by the EDGAR inventory.

### 3.5 Sensitivity to different model assumptions

A set of additional HFC-125 experiments was conducted by a subset of models to analyse the sensitivity to different

assumptions and identify possible reasons for the model-to-model differences (Table 2). A first test conducted by all models was an experiment for HFC-125 similar to M1 but using a flat, non-informative prior (FLAT), which had one emission value over land and one over ocean, to test the ability of the models to reconstruct the spatial distribution of emissions with no corresponding prior information. In this experiment, the uncertainty for the domain total emissions was set to 100%. Other experiments included tests with doubled (U200%) and halved (U50%) prior uncertainties conducted by NILU and UKMO, a

test with no optimization of the baseline conducted by EMPA2 (NOBLOPT), and tests with daily mean (DMEAN) and one

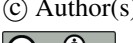



single observation per day (ONEOBS) instead of 3-hourly observations conducted by EMPA to mimic the sampling of NILU and UKMO.

The estimates with a flat prior (Figure 13, panel b) are similar to those with the spatially explicit prior (panel a) for most countries well covered by the footprint of the three measurement stations, notably for DE, IT, FR, UK, and IR, suggesting

that the model ensemble provides a robust estimate for these countries that is mainly informed by the measurements rather than the prior. This is less true for the individual models as shown in Table 3, which summarizes the results of all experiments for the largest well-covered countries in the domain. For countries in the east and northeast of the domain (SW+FI+BALT, NO, PO+CZ+SV), which are poorly 'seen' by the three sites, the median posterior remains close to the prior, and the posterior differences between experiments FLAT and M1 resemble the prior differences. For ES+PT both

priors are too low, but starting from a higher prior (experiment FLAT) results in an even higher posterior, especially in EMPA2 and UKMO.

Comparing the range of individual model estimates (Table 3 and uncertainty bars in Figure 13) suggests that model-to-model differences were of similar magnitude in experiments FLAT and M1 despite a more uniform setup in FLAT with an agreed total uncertainty. The differences thus appear to be mainly caused by the many other choices such as spatial correlations of

the prior, grid structure, background treatment, magnitude and correlation structure of the observation uncertainties, and transport model.

Some further insight is provided by the other sensitivity simulations: Decreasing or increasing the prior uncertainties by a factor of two relative to M1 changed the country estimates by only about 10% or less (Table 3). An exception is ES+PT where the results depended strongly on the prior uncertainty, which is a clear indication that the emissions from the Iberian

countries are not well constrained by the current observation network. Switching off the baseline optimization in EMPA2 to mimic the setup of NILU increased the emissions in all countries between +6% (DE) and up to +19% (FR, ES+PT). This indicates that with optimization the baseline in EMPA2 tended to be corrected upward and that without optimization this had to be compensated by higher emissions. However, the effect will strongly depend on the way the baseline is specified a priori. Switching off the baseline optimization did not bring EMPA2 closer to NILU.

Finally, the influence of different sampling and averaging of the observations was tested with the EMPA system in experiments DMEAN and ONEOBS to mimic the sampling of NILU and UKMO, respectively. Note that for experiment DMEAN the model-data mismatch uncertainty was reduced to respect the requirement of a $\chi^2$ value close to the number of observations (Brunner et al., 2006). The results for DE and IT changed only little but they changed substantially for FR, UK and ES+PT. With daily averaged instead of 3-hourly observations the estimate for FR increased by 17%, and with one

observation per day decreased by 22%, the latter being closer to the prior. For the UK, however, the opposite effect is seen, with daily means reducing (-13%) and one-observation-per-day increasing (+31%) the estimate relative to M1. The results for the UK are dominated by observations from the station Mace Head. At this site, the mean diurnal cycle of the differences between FLEXPART simulated and observed concentrations exhibits negative differences (-0.07 ppt) in the afternoon but positive differences (0.02-0.05 ppt) during the rest of the day. When using only afternoon observations as in experiment

ONEOBS and as used by UKMO, the EMPA system thus requires higher emissions to compensate for the negative bias compared to when all data are used. Both experiments suggest a considerable impact of the choice of observations, which is in contrast to previous findings of Brunner et al. (2012), who made a similar sensitivity experiment and found only a relatively small influence. Except for the UK, the estimates of experiment DMEAN were always higher than those of experiment ONEOBS, consistent with NILU being generally higher than UKMO. Some of the differences between the

model results are thus likely attributable to the specific selection and aggregation of the observation data.

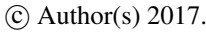



## 4 Conclusions

For the first time, four independent regional inversion systems for synthetic greenhouse gas emissions have been applied in well-controlled model experiments to compare the systems and to analyse the performance of the ensemble. Emissions of the two most important halocarbons in terms of ($CO_2$-eq.) greenhouse gas emissions in Europe, HFC-125 and HFC-134a, as well

as $SF_6$ were estimated for the year 2011. The four model systems, referred to as EMPA, EMPA2, NILU, and UKMO, differed in terms of Lagrangian transport model (3 x FLEXPART with ECMWF IFS meteorology, 1 x NAME with UKMO meteorology) and inversion method (3 x Bayesian inversion, 1 x extended Kalman Filter). The inversion systems used the same observation time series and a priori emission fields but differed in a number of other aspects such as the amplitude and correlation structure of the prior and observation uncertainty covariance matrices, the treatment of background mole

fractions, the inversion grid and resolution, and the averaging or subsampling of observations, in order to preserve the characteristics of the individual approaches as used in previous studies as much as possible.

All systems were able to reproduce the measurement time series well to very well. Pearson's correlation coefficients for the prior simulations were typically in the range 0.6-0.7 at Jungfraujoch, 0.8-0.9 at Mace Head, and 0.5-0.7 at Monte Cimone. Correlation coefficients for the posterior time series were about 0.05 to 0.1 better and bias-corrected RMSE were typically

reduced by 10 to 40% with the exception of HFC-134a at Monte Cimone, where the reduction was only between 2 and 5% in all systems. The transport model NAME was less successful than FLEXPART in reproducing the measurements at the two mountain sites JFJ and CMN but showed comparable performance at MHD.

The comparison of gridded emissions was complicated by the large differences in resolution and structure of the inversion grids: the number of grid elements optimized varied between 150 in the UKMO, 522 in EMPA2, 1083 in EMPA and 1140 in

the NILU system. UKMO, EMPA and EMPA2 had high grid resolution near the measurement sites and lower resolution at larger distance where the measurements were less sensitive, especially over eastern and south-eastern Europe and Scandinavia. The UKMO grid followed the country borders to simplify emission attribution to individual countries.

For HFC-125, all inversion systems estimated higher posterior emissions compared to the EDGAR v4.2 prior for the Iberian Peninsula and most of Italy except northern Italy. The models also tended towards higher posterior emissions over Ireland

and southwestern UK but lower emissions over the eastern and northern parts of UK. A unique feature of the NILU system was a band of positive posterior – prior differences extending from Germany towards the Baltic countries. For HFC-134a, the patterns of changes were similar but showed more negative posterior – prior differences (e.g., over the Benelux countries and the UK). For $SF_6$, all models simulated the highest emissions over Germany though much reduced with respect to the EDGAR v4.2 prior. In contrast to Germany, $SF_6$ emissions for Italy and France were higher than the prior.

Overall, NILU and EMPA tended to retrieve higher emissions than UKMO and EMPA2. For all three gases, NILU had the highest total domain emissions and EMPA2 the lowest. These results are related to two main factors: First, EMPA and NILU were the only systems considering spatial correlations in the prior resulting in a smaller number of degrees of freedom and a correspondingly stronger influence of the observations on the posterior emissions. Second, NILU was the only system not applying a correction to the background in order to avoid cross-talk between the optimization of the emissions and the

background. A sensitivity experiment for HFC-125 with no background adjustment conducted by EMPA2 indeed resulted in higher emissions though not reaching the levels of NILU.

The patterns of uncertainty reductions differed strongly: NILU and EMPA had rather smooth reductions whereas the patterns of EMPA2 and UKMO were more scattered due to the absence of spatial correlations in the prior uncertainties. NILU assumed large and rather uniform (absolute) prior uncertainties and, as a result, found the largest uncertainty reductions.




UKMO also had large prior uncertainties but much smaller reductions due to their assumption of large observation uncertainties.

Gridded emissions were aggregated to individual countries to analyse the consistency between the models and to compare the results against country totals officially reported to the UNFCCC (reported in 2013 for the year 2011) and to the EDGAR

v4.2 prior (representing 2008). The rather coarse inversion grids were a non-negligible source of uncertainty (typically between 1 and 6%) when aggregating the emissions to individual countries. The overall magnitude of the emissions and the attribution to different countries such as the dominant role of Germany for $SF_6$ emissions were quite consistent with the UNFCCC estimates. However, the estimates of the individual models varied considerably. Considering all three gases and the largest countries, the scatter was smallest for the UK ($1\sigma$ standard deviation of 3-11%), followed by France (8-15%),

Germany (19-22%), Italy (12-31%), and Spain+Portugal (24-30%). The individual models often did not overlap within the range of the combined uncertainties suggesting that the analytical uncertainties are a poor representation of the true uncertainties, which are rather dominated by parametric and structural uncertainties.

The ensemble median agreed very well with the UNFCCC estimates for HFC-134a for most countries, better than any single model. As also found in previous studies, emissions of HFC-134a reported to UNFCCC by the UK appear to be about a

factor two too high. A similar conclusion may be drawn for the group Poland+Czech Republic+Slovakia though with less confidence due to the limited coverage of these countries by the current observation network. In terms of HFC-125 emissions the largest discrepancies from UNFCCC values were found for Spain+Portugal and for Ireland, with model medians 4.9 times and 9.5 times higher, respectively. Interestingly, for the same countries the estimates for HFC-134a were highly consistent with the models, providing further evidence that the reported HFC-125 emissions are too low. Consistent

with the UNFCCC reports, the models identified Germany as the highest national emitter of $SF_6$ in Europe. The model estimates for Germany agreed well with the UNFCCC numbers but were a factor 2 to 3 higher for Italy, France and Spain+Portugal.

The current network of three routine monitoring sites for synthetic greenhouse gases in Europe is only able to constrain the broad spatial patterns of their emissions, such as the concentration of $SF_6$ emissions on Germany as opposed to the more

uniform distribution of emissions of HFC-125 and HFC-134a. The network has the potential to identify significant shortcomings in the nationally reported emissions but a denser network would be needed for a more accurate assignment to individual countries. Model-to-model differences were often very large whereas the model median appears to have significant skill as judged from the comparison with reported HFC-134a emissions, which are considered to be relatively well known. The sensitivity experiments were not sufficient to fully disclose the origin of the model-to-model differences,

but factors such as subsampling of observations, background treatment, and magnitude and correlation structure of the prior uncertainties were identified as playing an important role. Further work will be needed, for example by testing the model's internal consistency using a $\chi^2$ test, and by separating model transport from other uncertainties, to build trust in the inverse modelling systems.

**Acknowledgements**

This study was funded by the European Commission's Seventh Framework Programme project InGOS (grant agreement no. 284274). Measurements at Jungfraujoch are supported by the Swiss Federal Office for the Environment (FOEN) through the project HALCLIM and by the International Foundation High Altitude Research Stations Jungfraujoch and Gornergrat (HFSJG). Measurements at Mace Head are supported by the Department of Energy & Climate Change (DECC, UK)

(contract GA0201 with the University of Bristol). The O. Vittori station Monte Cimone is supported by the National Research Council of Italy.



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



**Table 1:** Overview of inversion systems

| Characteristic | EMPA | EMPA2 | NILU | UKMO |
|---|---|---|---|---|
| Approach | Extended Kalman Filter (ExKF) | Bayesian | Bayesian | Bayesian |
| Transport model | FLEXPART | FLEXPART | FLEXPART | NAME |
| Meteorology | ECMWF analyses 0.2°x0.2°, 3hrly | ECMWF analyses 0.2°x0.2°, 3hrly | ECMWF analyses 0.2°x0.2°, 3hrly | UKMO analyses 0.352° x 0.234°, 3hrly |
| Computational domain | Nested, global | Nested, global | Nested, global | 45°W - 40°E, 25°N - 80°N |
| Inversion grid | 0.1°x0.1°min., reduced according to residence time | 0.1°x0.1°min., reduced according to residence time | 1°x1° over land, reduced over ocean and far eastern boundary | 0.352° x 0.234° min., reduced acc. to residence time and within country boundaries |
| State vector length (e=emiss., b=backg., o=other) | $1083e + 3b + 6o$ | 522 e + 84 b (405 e + 56 b for M3) | 1140e | 150e + 11 b |
| Assimilation time resolution | 3-hourly means | 3-hourly means | Daily means | 3-hourly means once per day |
| Spatial correlation of prior | 500 km | None | 200 km over land 1000 km over sea | None |
| Backwards mode run time | 5 days | 5 days | 10 days | 19 days |
| Prior background mole factions | None, continuously estimated by ExKF | 60-day REBS window, biweekly reference points | See Thompson and Stohl (2014) and description below. | Mace Head baseline for all sites, see Manning et al. (2011) |
| Temporal correlation of observation error | Red-noise Kalman filter | None | None, assumed negligible for daily means | None, assumed negligible with one value per day |
| Key references | Brunner et al., 2012 | Stohl et al., 2009, Vollmer et al., 2009 | Thompson and Stohl, 2014 | Manning et al., 2011 |





**Table 2:** Main (M1-M3) and sensitivity inversion experiments

| ID | Gas | Prior inventory | Description | Groups |
|---|---|---|---|---|
| M1 | HFC-125 | EDGARv4.2 2008 | Reference inversion for HFC-125 for 2011 | All |
| M2 | HFC-134a | EDGARv4.2 2008 | Reference inversion for HFC-134a for 2011 | All |
| M3 | $SF_6$ | EDGARv4.2 2008 | Reference inversion for $SF_6$ for 2011 | All |
| FLAT | HFC-125 | Uniform prior[1] | Spatially uniform prior instead of EDGAR | All |
| U50% | HFC-125 | EDGARv4.2 2008 | Prior uncertainty reduced by factor 2 | UKMO, NILU |
| U200% | HFC-125 | EDGARv4.2 2008 | Prior uncertainty increased by factor 2 | UKMO, NILU |
| NOBLOPT | HFC-125 | EDGARv4.2 2008 | No baseline optimization | EMPA2 |
| DMEAN | HFC-125 | EDGARv4.2 2008 | Daily means instead of 3-hourly | EMPA |
| ONEOBS | HFC-125 | EDGARv4.2 2008 | One instead of eight observations per day | EMPA |

[1] One value over land and one value over sea

**Table 3: E**missions of HFC-125 in the main experiment M1 and the different sensitivity experiments for major countries in western Europe. UNFCCC refers to the 2011 emissions according to the country reports submitted to UNFCCC in 2013. EDGAR v4.2 refers to 2008 emissions according to the gridding method applied by EMPA2. Uncertainties are shown as ±1σ estimates.

| Exp. ID | Model/Inventory | DE | | IT | | FR | | UK | | ES+PT | |
|---|---|---|---|---|---|---|---|---|---|---|---|
| | | (Mg yr⁻¹) | | (Mg yr⁻¹) | | (Mg yr⁻¹) | | (Mg yr⁻¹) | | (Mg yr⁻¹) | |
| | *UNFCCC 2011* | *548* | | *1169* | | *1234* | | *1061* | | *390* | |
| | *EDGAR v4.2 2008* | *1232* | | *801* | | *1001* | | *793* | | *491* | |
| M1 | EMPA | 1094 | ± 237 | 2138 | ± 240 | 1483 | ± 180 | 918 | ± 144 | 2599 | ± 353 |
| | EMPA2 | 721 | ± 196 | 1212 | ± 73 | 787 | ± 100 | 812 | ± 64 | 1076 | ± 121 |
| | NILU | 2078 | ± 22 | 1039 | ± 7 | 1195 | ± 13 | 758 | ± 13 | 2849 | ± 17 |
| | UKMO | 1568 | ± 327 | 1021 | ± 102 | 919 | ± 123 | 702 | ± 235 | 1218 | ± 136 |
| | *Median* | *1331* | | *1125* | | *1057* | | *785* | | *1909* | |
| | *Range (min-max)* | *721-2078* | | *1021-2138* | | *787-1483* | | *702-918* | | *1076-2849* | |
| FLAT | EMPA | 1016 | ± 354 | 1522 | ± 285 | 1929 | ± 295 | 1172 | ± 273 | 2713 | ± 537 |
| | EMPA2 | 772 | ± 142 | 1302 | ± 149 | 1067 | ± 134 | 651 | ± 94 | 1769 | ± 245 |
| | NILU | 1956 | ± 20 | 736 | ± 17 | 1037 | ± 17 | 535 | ± 16 | 2928 | ± 29 |
| | UKMO | 1586 | ± 946 | 1115 | ± 276 | 1276 | ± 298 | 737 | ± 440 | 3009 | ± 499 |
| | *Median* | *1301* | | *1209* | | *1172* | | *694* | | *2820* | |
| | *Range (min-max)* | *772-1956* | | *736-1522* | | *1037-1929* | | *535-1172* | | *1769-2928* | |
| U50% | NILU | 2151 | ± 21 | 1055 | ± 6 | 1292 | ± 10 | 766 | ± 10 | 2372 | ± 14 |
| | UKMO | 1539 | ± 195 | 910 | ± 72 | 824 | ± 98 | 797 | ± 145 | 899 | ± 91 |
| U200% | NILU | 1936 | ± 21 | 1033 | ± 10 | 1030 | ± 14 | 746 | ± 14 | 3426 | ± 19 |
| | UKMO[1] | 1422 | ± 545 | 999 | ± 165 | 1066 | ± 164 | 530 | ± 330 | 1739 | ± 208 |
| NOBLOPT | EMPA2 | 770 | ± 196 | 1330 | ± 71 | 937 | ± 98 | 926 | ± 64 | 1284 | ± 118 |
| DMEAN | EMPA | 1123 | ± 471 | 2192 | ± 500 | 1739 | ± 399 | 797 | ± 271 | 2582 | ± 780 |
| ONEOBS | EMPA | 1068 | ± 491 | 2015 | ± 559 | 1138 | ± 337 | 1209 | ± 460 | 1655 | ± 604 |
| | *Median* | *1488* | | *1055* | | *1066* | | *797* | | *1740* | |
| | *Range (min-max)* | *770-2151* | | *910-2192* | | *824-1739* | | *530-1209* | | *899-3426* | |

[1] Uncertainty increased by 250% rather than 200%





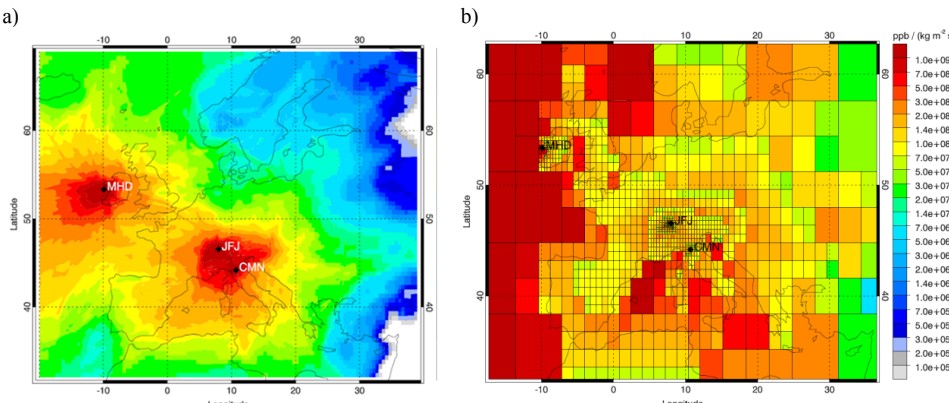

**Figure 1:** Total surface sensitivity in units of [ppb/(kg m$^{-2}$ s$^{-1}$)] for (a) the original 0.1°x0.1°grid and (b) the reduced grid.





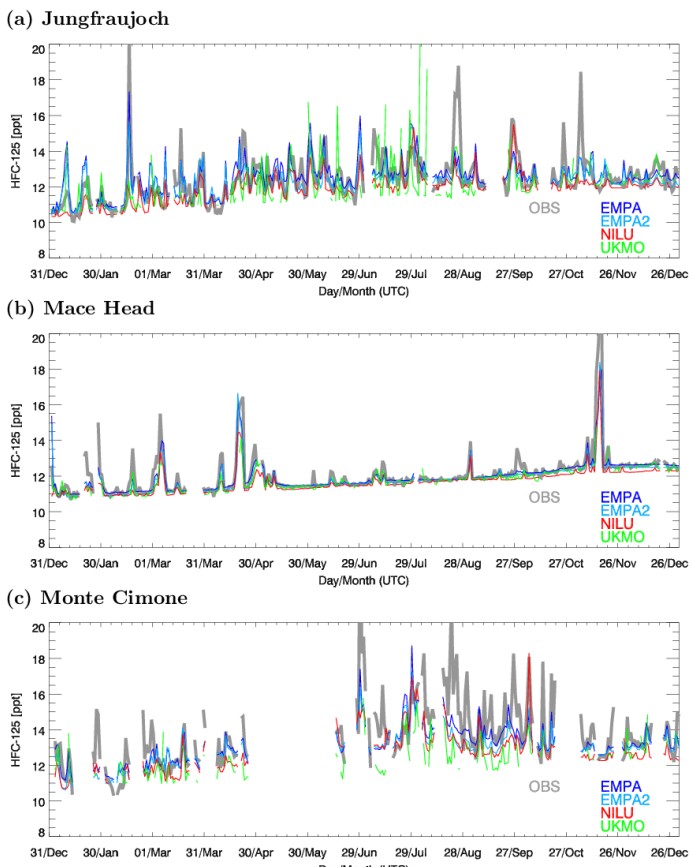

**Figure 2:** Prior simulated HFC-125 mole fractions (colour lines) overlaid over observations (thick grey line) at the three sites Jungfraujoch, Mace Head and Monte Cimone.





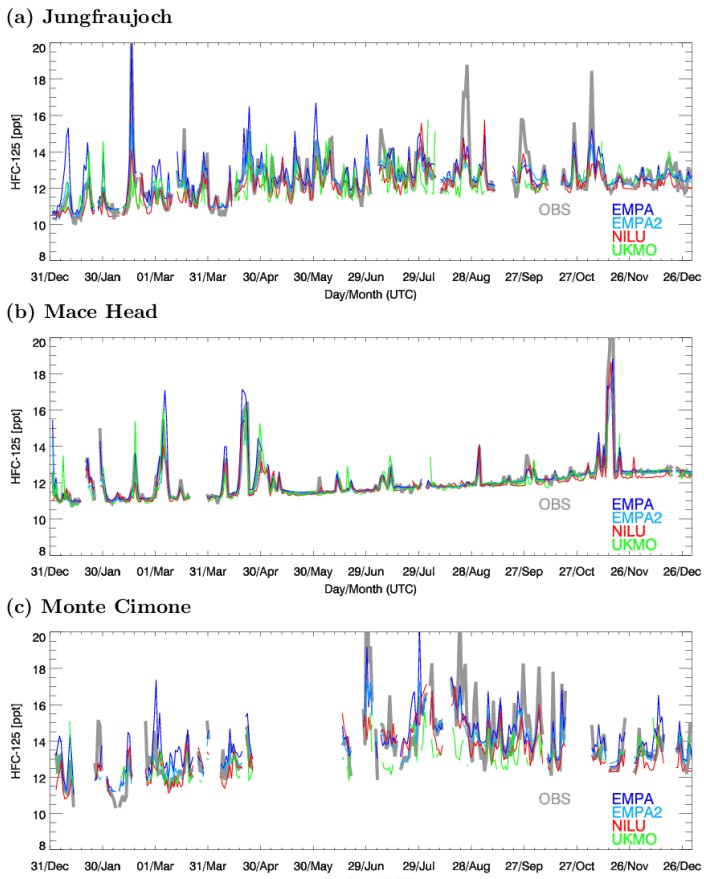

**Figure 3:** Same as Figure 2 but for posterior simulations.





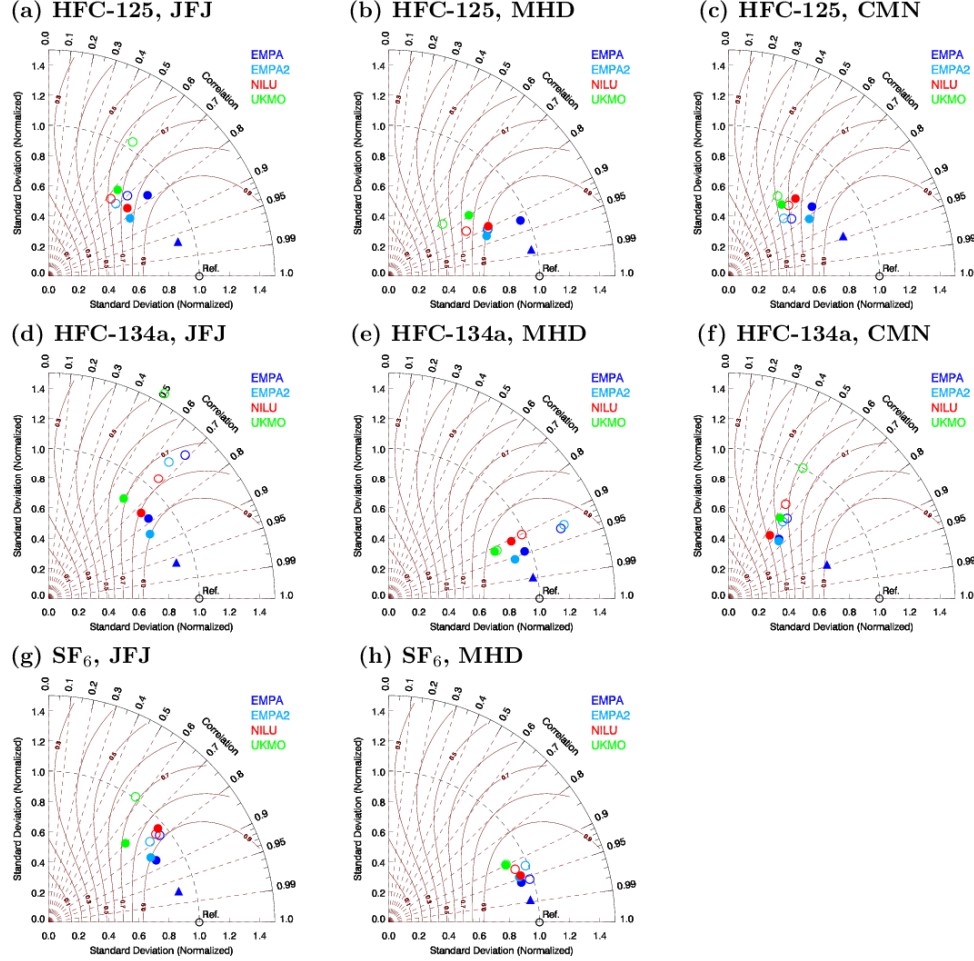

**Figure 4:** Taylor diagrams of model performance for the simulated prior (open circles) and posterior (filled circles) mole fraction time series. The filled blue triangle for EMPA indicates the performance when including an AR(1) autocorrelation term in the Kalman filter. The linear distance from the reference point (Ref.) is proportional to the centred (bias corrected) root mean square error (RMSE). The angle of rotation with respect to the vertical axis corresponds to the Pearson correlation coefficient R.





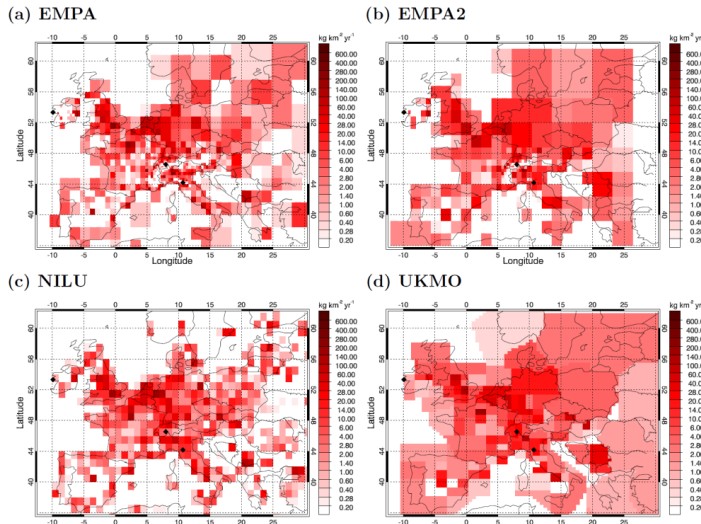

**Figure 5:** Prior emissions of HFC-134a as represented in the four inversion systems.

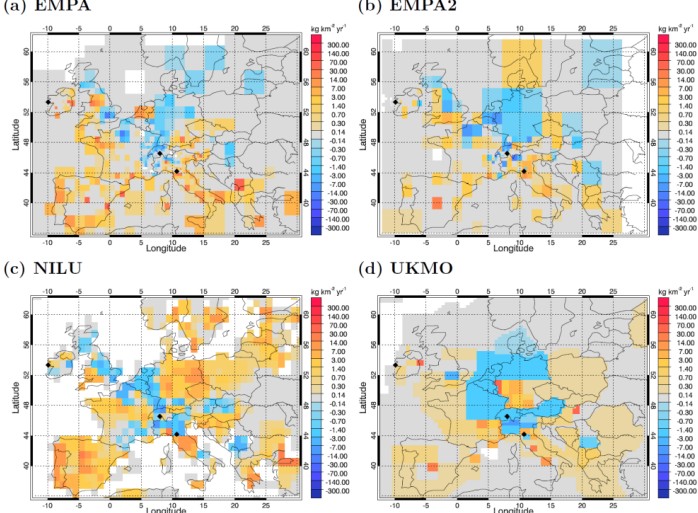

**Figure 6:** Posterior – prior HFC-125 emission differences (experiment M1).



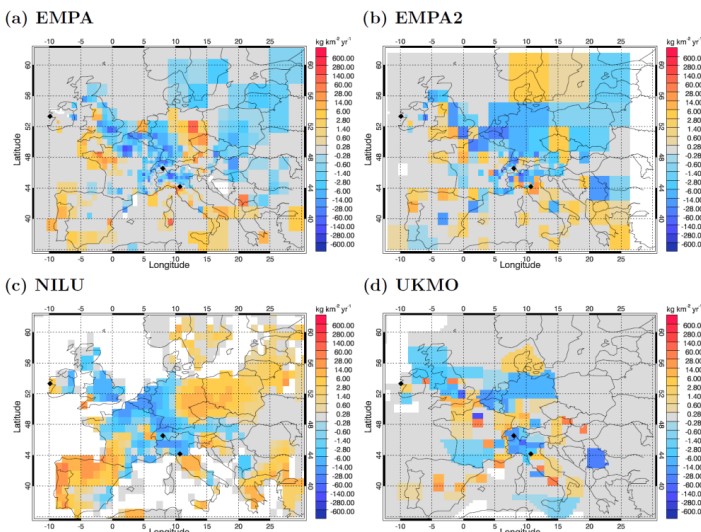

**Figure 7:** Posterior – prior HFC-134a emission differences (experiment M2).

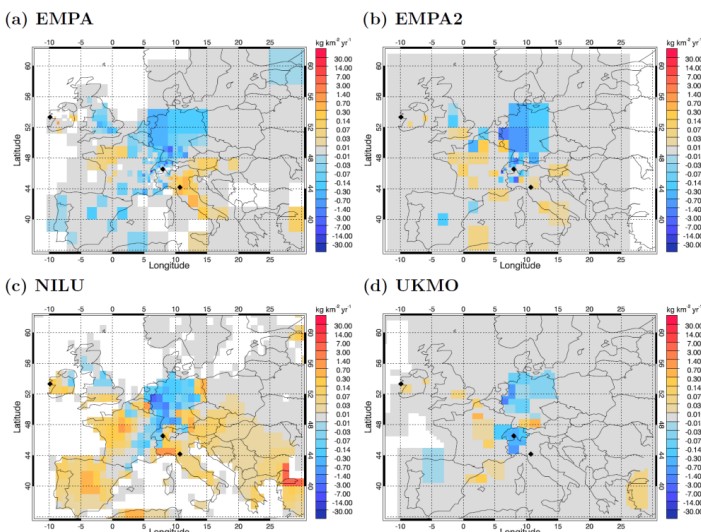

**Figure 8:** Posterior – prior SF$_6$ emission differences (experiment M3).



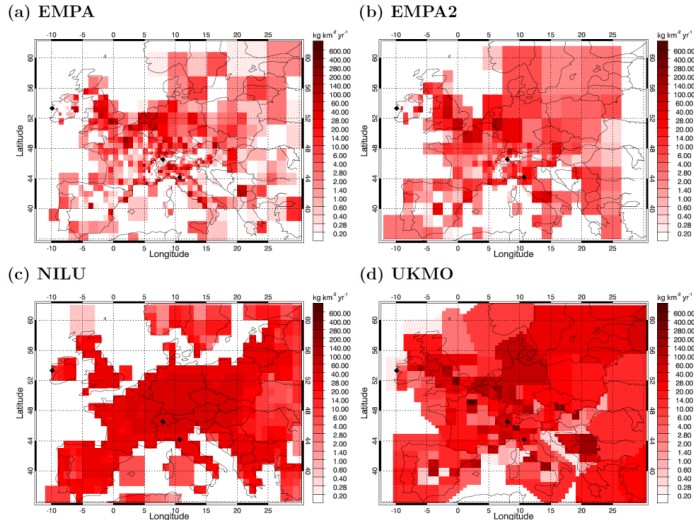

**Figure 9:** Uncertainty of prior HFC-134a emissions (experiment M2).

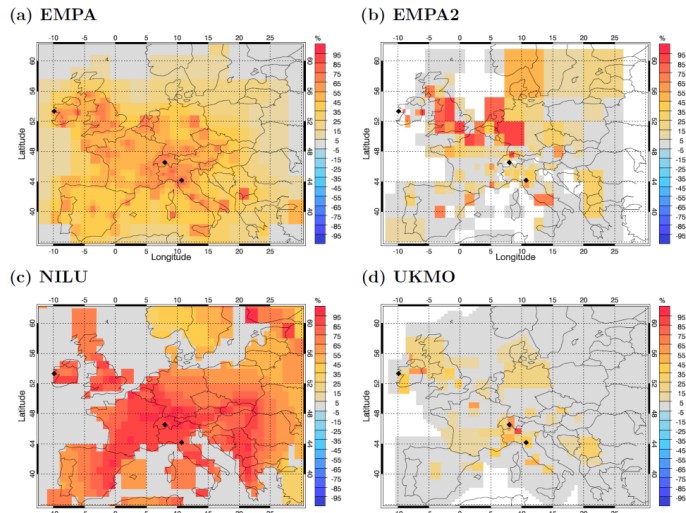

**Figure 10:** Uncertainty reduction $(1-u_{post}/u_{prior})$ in % for HFC-134a (experiment M2). For EMPA, the reduction is shown in terms of reduction of relative uncertainties $[1-(u_{post}/x_{post})/(u_{prior}/x_{prior})]$.




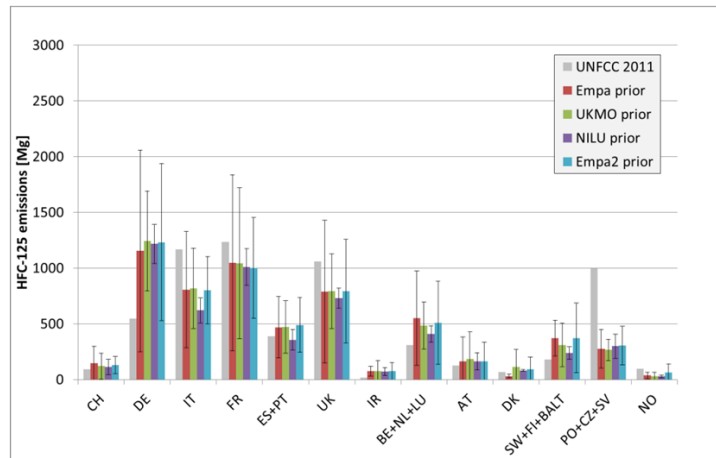

**Figure 11:** Country-aggregated prior emissions of HFC-125 (experiment M1). Country codes following ISO2 conventions except for BALT = Baltic countries (Estonia, Latvia and Lithuania).

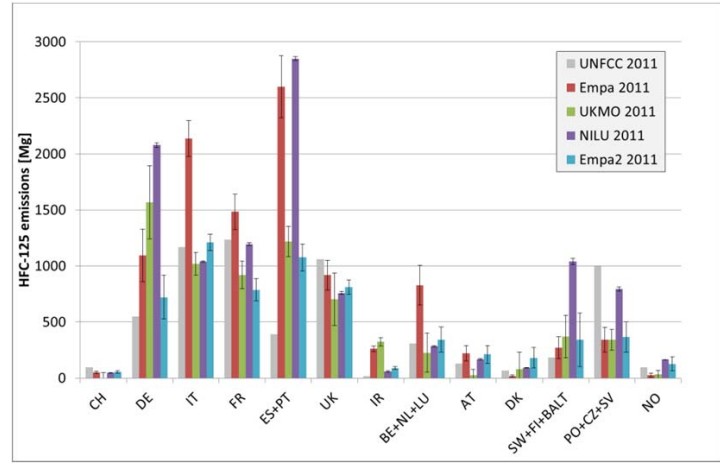

5    **Figure 12:** Country-aggregated posterior emissions of HFC-125 (experiment M1).





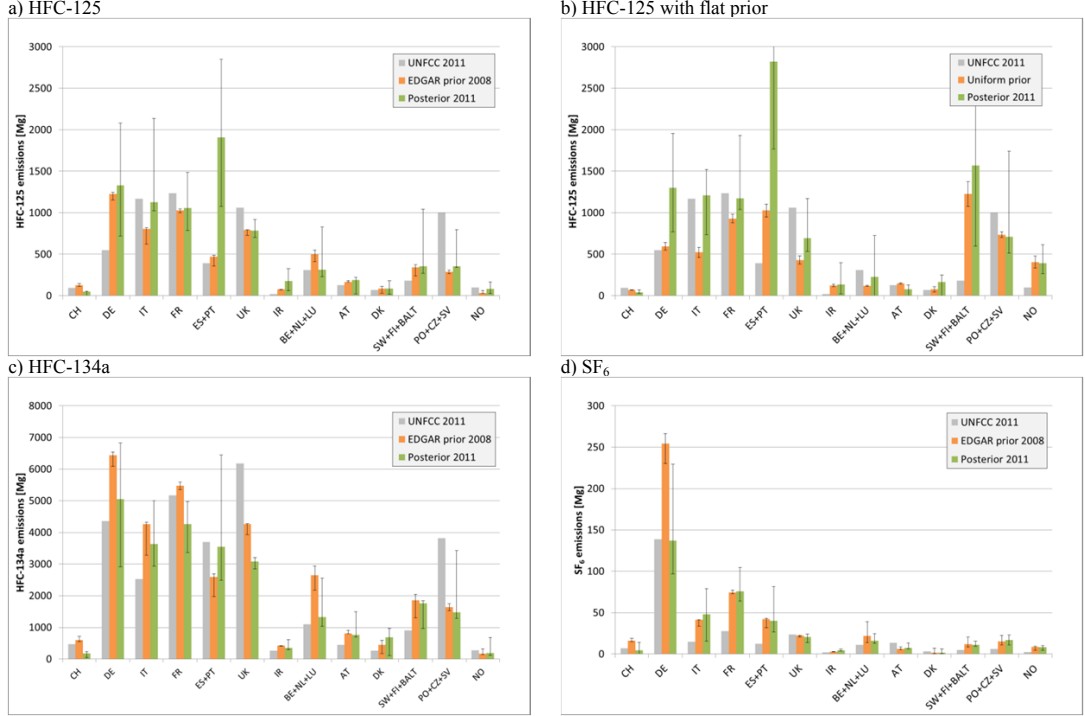

**Figure 13:** Median country-aggregated posterior emissions for a) HFC-125 (experiment M1), b) HFC-125 with flat prior (experiment FLAT), c) HFC-134a (experiment M2), d) SF$_6$ (experiment M3). Uncertainties bars denote the range between minimum and maximum estimate of the four models.





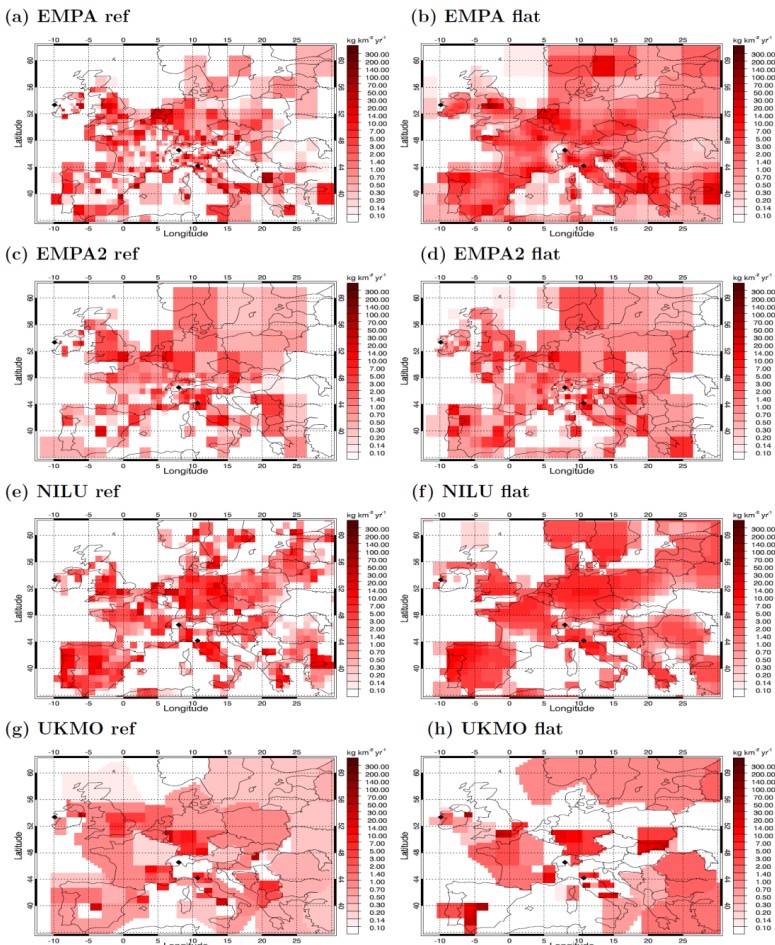

**Figure 14:** Posterior emissions of HFC-125 for the reference experiment M1 (left column) and the experiment with flat prior (right column).