# Peer review of "Comparison of four inverse modelling systems applied to the estimation of HFC-125, HFC-134a and $SF_6$ emissions over Europe"

_Atmospheric Chemistry and Physics, 2017_

## Referee Comment (RC1) · Anonymous Referee #2 · 13 Mar 2017

The paper by Brunner et al. reports on the inverse estimation of hydrofluorocarbon (HFC) and SF6 emissions in Europe exploiting the respective atmospheric concentrations variability measured at three monitoring sites. Four state-of-the-art inverse modelling systems are compared with respect to their performance in modelling the observed concentrations and with respect to their consistency in inverting surface fluxes. The model-median emissions are then discussed for country-wide regions and compared to reported bottom-up inventories.

The study contributes important insight into model performance for providing top-down constraints on anthropogenic greenhouse gas emissions. Comparing the top-down emission estimates to bottom-up inventories yields significant discrepancies for some

European countries which indicates that more research is required to consolidate emission reporting and verification.

The paper is very well written, the methods are state-of-the-art, and the topic is interesting for the atmospheric sciences community. Therefore, I recommend publication after addressing my few comments below.

The conclusions on country-wide emissions appear somewhat unconsolidated given that the model-to-model differences are as large as the estimated emissions for some countries (e.g. Figure 12). While I accept the approach to use model versions that are as close as possible to the respective production settings, it is quite unsatisfying that the reasons for these model differences are essentially unresolved. In that context, I am also not convinced by using the model median value (of only 4 models). I would suggest making abstract and conclusions somewhat humble by adding some more discussion on how the discrepancies between bottom-up and top-down emissions compare to model differences.

A detail that came to my attention is that the release height for the particles at Jungfraujoch was adjusted for the NAME model to match the FLEXPART footprints. Essentially, this adjustment appears arbitrary and contradicts the general philosophy to use production settings for each model. If the adjustment was not made (transport induced) model differences would be even larger. So, given that (at least one of) the transport models are not able to correctly model transport at the mountain sites, how confident are you with respect to your overall conclusions?

P2,L24: regulated reported -> reported

P9,L6 and following: Occasionally, I got confused by the naming conventions. I would suggest using NAME and FLEXPART when referring to transport issues and the others names when referring to the entire modelling systems: P9,L6: UKMO -> NAME, P9,L13: NAME->UKMO, check other places.

---

## Referee Comment (RC2) · Anonymous Referee #1 · 14 Mar 2017

This is a clearly written, informative, and useful paper that adds significantly to our understanding of regional emissions of long-lived trace gases derived from atmospheric data. I had only a few thoughts on clarifications and adjustments to improve the paper:

On the gridded emissions. Some issues could be addressed to make things more robust and clear. It is indeed striking the visual differences in different approaches to gridding the EDGAR emissions in Figure 5. It would have helped me if you had mentioned in this section the spatial distribution of the native EDGAR inventory estimates and, how consistent country totals are after this gridding by the different methods (shown in Figure 11). Given the rather significant and arbitrary variations in the priors, a discussion of emission updates (Figures 6-8) becomes one that is related to two factors: the

arbitrary errors in the priors because of the imperfect gridding process, and differences in model performance. At this point in the text only the second influence is considered, though it seems necessary to consider how the first factor is influencing the results too. (In other words, if all the models performed exactly the same in their inversion, there would still be substantially different updates apparent in Figure 6-8 because of the different gridding errors associated with the prior.) The better discussion of these issues comes later in the text in the comparison of figures 11 and 12, in my opinion. The authors might consider shortening or revising this earlier section.

Regarding the apparent large differences in adjustments by the different models despite the reasonable similarity in posterior mole fraction time series generated by these models: It would seem that these aren't directly relatable unless you consider the sum of the fluxes shown in Figures 5 and 6, given that the posterior mixing ratios are from the sum of the prior emissions plus adjustments. Given the large apparent differences in the priors because of the different gridding approaches, this seems important to consider.

On background levels. Since the approach for deriving background mole fractions taken by NILU is unique because it involves a subtraction related to the calculated influence of regional emissions on the observations deemed to represent background, it would seem reasonable to suggest that this subtraction might be causing the lower background mole fractions it derives. Is it not fairly easy to determine if this is the source of the offset?

Another minor issue, with regard to backgrounds for the approaches by EMPA2. The REBS approach is mentioned and an optimization is also indicated. Details about the optimization are lacking. Was the optimization applied to the REBS results? And how was that process constrained? Does the text mentioning that "the background is then allowed to evolve slowly with time" mean that it was just another optimized parameter in the inversion who's only constraint was low-frequency variation?

On section 3.3., uncertainty reductions. The authors seem to succeed in showing evidence refuting the initial statement that this is "a useful diagnostic" since the magnitudes seem primarily dependent on what is assumed as the uncertainty on the prior! In looking for robust conclusions from this section, there is one that I struggle to reconcile: How can uncertainty reductions expressed relative to absolute emission magnitudes be larger for those regions with higher emissions? Some explanation would be helpful here, since it seems not an expected or straightforward conclusion.

Details: Sentence two of abstract, consider adding a word: "but *emissions* have large GWPs and are, therefore..." Also, in the abstract the discrepancy in HFC-125 emissions estimated for the Iberian peninsula is the first point made in the comparison of results vs the UNFCCC inventory emissions, yet the main text mentions that "emissions from the Iberian countries are not well constrained by the current observation network." Some modifications to the abstract seem necessary.

Define "standard deviation (normalized)" in the caption of the figure showing Taylor diagrams. I presume it is the ratio between the observed vs posterior calculated mole fractions—this should be mentioned if true. Any de-trending applied to the results over the year, or is it just the s.d. of the annual data record considered together?

Figure 1 caption, mention that the reduced grid is only associated with the EMPA simulations, if true.
* * *

---

## Referee Comment (RC3) · Anonymous Referee #3 · 6 Apr 2017

The authors used four inverse models to estimate European emissions of HFC-134a, HFC-125 and $SF_6$ for the year 2011. All systems used measurements from Jungfraujoch, Mace Head, and Monte Cimone. The paper is well written and provides interesting insights. I think the main problem of the paper was that the differences in the choices, such as spatial correlations of the prior and background treatment, had a quite substantial impact on differences among the models. What was the reason that those were not controlled? If they were better controlled, maybe we could have had more insights on which models are doing better and what we might do to improve the emissions estimation through inverse modeling. Below are some other comments and questions I had and I would recommend publication after they are addressed.

For Figure 1, is this the sensitivity created using FLEXPART or NAME? I would also assume that the sensitivity is quite different depending on the month. Which month is this? And is this the monthly mean?

It was a little unclear why NAME needed such a high release height at Jungfraujoch. If the point of the paper is to better understand the differences among the four inversion systems, I find it puzzling that the authors would modify to make the model footprint sensitivities comparable to each other.

I had a hard time understanding how the emissions were created following the country outlines. What was the means used to split the EDGAR grid to country outlines? Also, because the prior emissions are so different, I find it more informative if the Fig. 6 was not comparing between prior and posterior but EDGAR and posterior.

Why did EMPA2 use the uncertainty set uniformly to 137%? This seemed a little strange and was curious for the reason behind this specific value.

One of the explanations for why UK's estimated emissions are much higher than what is reported to UNFCCC, the authors mention the use of an assumed high loss rate of HFC-134a from car air conditioning systems in the UK. Why is this only in the UK and how different is the loss rate among the countries? Is a similar explanation possible for overestimation and/or underestimation for different species?

Backwards mode time differ substantially among the models and I would have expected UKMO to have larger difference between prior and posterior away from the measurement sites, compared to the other model systems that have shorter time span. Why is it that UKMO shows almost no difference between the two farther away from the measurement sites?

Minor comments
1. Sometimes authors state the country by name and sometimes by the ISO2 convention country code. It is a little confusing to me and so I would suggest to be consistent and I would appreciate if there was a table listing the country names with ISO2 code if the authors want to use the codes.

2. P. 11 l. 4 "An important question is the context … is the question" → delete the second "the question" in the sentence to make it "… Paris Agreement is, how suitable is…"
3. I am not quite sure what 0.1°x0.1°min means in Table 1.
4. "reduced acc. to" → "reduced according to" in Table 1 for UKMO
5. State vector length is mentioned in Table 1 but was not explained in the text at all. Can this be clarified in terms of how this is used in the equation and why the equations look so different depending on the system?
6. How is the EDGAR prior uncertainty determined in Figure 13? I find that to be a little misleading, since I do not think EDGAR provides such a value.
7. Figure 14 is very difficult to see – maybe a different color scheme would work better.

---

## Author Comment (AC1) · 14 Jun 2017

**Reply to referee#1**

The authors would like to thank the referee for the careful review and the helpful comments. In the following, the reviewer's comments will be in **bold** font, and the responses will be in plain font, with suggested new text in *italics*.

**On the gridded emissions. Some issues could be addressed to make things more robust and clear. It is indeed striking the visual differences in different approaches to gridding the EDGAR emissions in Figure 5. It would have helped me if you had mentioned in this section the spatial distribution of the native EDGAR inventory estimates and, how consistent country totals are after this gridding by the different methods (shown in Figure 11). Given the rather significant and arbitrary variations in the priors, a discussion of emission updates (Figures 6-8) becomes one that is related to two factors: the arbitrary errors in the priors because of the imperfect gridding process, and differences in model performance. At this point in the text only the second influence is considered, though it seems necessary to consider how the first factor is influencing the results too. (In other words, if all the models performed exactly the same in their inversion, there would still be substantially different updates apparent in Figure 6-8 because of the different gridding errors associated with the prior.) The better discussion of these issues comes later in the text in the comparison of figures 11 and 12, in my opinion. The authors might consider shortening or revising this earlier section.**

Although the different grids have largely different resolutions and structure, all gridding algorithms are conserving the mass emitted in the original EDGAR v4.2 inventory. In that sense there are no "imperfections" or "gridding errors". Differences in the a priori emissions only occur for smaller spatial aggregates such as country totals, that do not perfectly align with the grid structure. We added this information as well as the spatial resolution of EDGAR v4.2. as follows:

*Although based on exactly the same EDGAR v4.2 inventory data, w**hich has a resolution of 0.1° x 0.1°**, the spatial aggregation to the different inversion grids leads to visually quite different distributions **despite the fact that all gridding algorithms are mass conserving, i.e. the emission from a coarse grid cell exactly corresponds to the sum of emissions from all finer EDGAR grid cells within that cell***.

In this section, we tried to focus on the broad spatial patterns, which should be much less sensitive to the specific grid configuration than the analysis of country totals.

**Regarding the apparent large differences in adjustments by the different models despite the reasonable similarity in posterior mole fraction time series generated by these models: It would seem that these aren't directly relatable unless you consider the sum of the fluxes shown in Figures 5 and 6, given that the posterior mixing ratios are from the sum of the prior emissions plus adjustments. Given the large apparent differences in the priors because of the different gridding approaches, this seems important to consider.**

The mole fraction simulated for a given measurement location and time is determined by the fluxes within its footprint plus background. Assuming that the footprints of the transport models are similar/identical (which is certainly true for the three FLEXPART systems), a similarity in the time series can be translated into the expectation that the spatial emission patterns are similar, too. We agree that the fluxes correspond to the sum of Figures 5 and 6, but since there are no biases in the priors due to the conservation of emissions in each grid cell (as explained above), we think that Figure 6 alone is sufficient to discuss the broad spatial patterns.

**On background levels. Since the approach for deriving background mole fractions taken by NILU is unique because it involves a subtraction related to the calculated influence of regional emissions on the observations deemed to represent background, it would seem reasonable to suggest that this subtraction might be causing the lower background mole fractions it derives. Is it not fairly easy to determine if this is the source of the offset?**

The procedure of NILU indeed leads to a lower background as compared to the other approaches. Combined with the fact that NILU does not adjust this background in the inversion, this likely leads to comparatively high emissions. We will conduct another simulation with the EMPA2 system mimicking this approach. We don't expect, however, that this will explain all differences, because the difference between background with and without correction for regional influence is expected to be small. Nevertheless, this is a very valid suggestion that will be included as an additional sensitivity test in the revised manuscript.

**Another minor issue, with regard to backgrounds for the approaches by EMPA2. The REBS approach is mentioned and an optimization is also indicated. Details about the optimization are lacking. Was the optimization applied to the REBS results? And how was that process constrained? Does the text mentioning that "the background is then allowed to evolve slowly with time" mean that it was just another optimized parameter in the inversion who's only constraint was low-frequency variation?**

Indeed, EMPA2 optimized the REBS background levels. We will make this clear in the text as follows:

*EMPA2 optimized the **REBS** background levels separately for each measurement site at selected reference points every 14 days. Background levels in between these reference points were linearly interpolated.*

And yes, in the EMPA system, which is sequentially applied to the data, the background level is another optimized parameter. It's update equation from one time step to the next follows the same equation (eq. 7) as the update for the emissions. The magnitude of the update uncertainty ($\varepsilon_k$) determines, how "slowly" the background is allowed to change from one time step to the next. We will add a reference to Equation (7) near the end of Sect. 2.3.

**On section 3.3., uncertainty reductions. The authors seem to succeed in showing evidence refuting the initial statement that this is "a useful diagnostic" since the magnitudes seem primarily dependent on what is assumed as the uncertainty on the prior! In looking for robust conclusions from this section, there is one that I struggle to reconcile: How can uncertainty reductions expressed relative to absolute emission magnitudes be larger for those regions with higher emissions? Some explanation would be helpful here, since it seems not an expected or straightforward conclusion.**

We fully agree that the discussion of uncertainty reductions is challenged by the fact that these strongly depend on the prior uncertainties. This issue is already addressed by the statement "Together with the different spatial uncertainty correlations, these differences have a marked effect on the resulting uncertainty reductions". We will better emphasize this issue already at the start of the section with a cautionary note:

*However, it should be noted that the uncertainty reduction depends on the magnitude and correlation structure of the prior uncertainties. Comparing the uncertainty reductions thus helps illustrating the effect of the different model choices.*

**Details: Sentence two of abstract, consider adding a word: "but *emissions* have large GWPs and are, therefore..." Also, in the abstract the discrepancy in HFC-125 emissions estimated for the Iberian peninsula is the first point made in the comparison of results vs the UNFCCC inventory emissions, yet the main text mentions that "emissions from the Iberian countries are not well constrained by the current observation network." Some modifications to the abstract seem necessary.**

We don't think that adding "emissions" would make the sentence more easily understandable. It is common practice to refer to the GWP of a gas rather than to the GWP of its emissions. Ultimately, it is the gas itself that has the properties leading to a high or low GWP.

It is true that emissions are not very well constrained for the Iberian Peninsula. Nevertheless, the fact that all models estimate much higher than reported emissions for HFC-125 but not for HFC-134a, is a strong indication that HFC-125 emissions are underreported. We will add a note of caution to the abstract:

*.. though with a large scatter between individual estimates.*

**Define "standard deviation (normalized)" in the caption of the figure showing Taylor diagrams. I presume it is the ratio between the observed vs posterior calculated mole fractions. Tthis should be mentioned if true. Any de-trending applied to the results over the year, or is it just the s.d. of the annual data record considered together?**

The word "normalized" refers to the fact that in a Taylor diagram the standard deviation of the simulated values is normalized by the standard deviation of the observations. A value of 1 indicates perfect agreement between the magnitude of scatter in the simulated and observed values. This information will be added to the caption.

**Figure 1 caption, mention that the reduced grid is only associated with the EMPA simulations,**

**if true.**

Correct, the figure caption was indeed lacking and will be change to:

*Annual mean surface sensitivity in units of [ppb/(kg m-2 s-1)] for (a) the original 0.1°x0.1°grid and (b) for the reduced grid of the FLEXPART-based model system EMPA.*

---

## Author Comment (AC2) · 14 Jun 2017

**Reply to referee#2**

The authors would like to thank anonymous referee #2 for the careful review and the helpful comments. In the following, the reviewer's comments will be in **bold** font, and the responses will be in plain font, with suggested new text in *italics*.

**The conclusions on country-wide emissions appear somewhat unconsolidated given that the model-to-model differences are as large as the estimated emissions for some countries (e.g. Figure 12). While I accept the approach to use model versions that are as close as possible to the respective production settings, it is quite unsatisfying that the reasons for these model differences are essentially unresolved. In that context, I am also not convinced by using the model median value (of only 4 models). I would suggest making abstract and conclusions somewhat humble by adding some more discussion on how the discrepancies between bottom-up and top-down emissions compare to model differences.**

Being humble in terms of conclusions about country scale emissions is a valid suggestion. In the abstract we will expand the sentence regarding the much higher simulated than reported HFC-125 emissions from Spain+Portugal with

*.. though with a large scatter between individual estimates*

and will add "country-scale" to the last sentence to read as follows:

*.. but a denser network would be needed for more reliable monitoring of **country-scale** emissions of these important greenhouse gases across Europe.*

In the conclusions section, the limitations of the inversions with respect to country emissions were already pointed out quite clearly, e.g. in the third last paragraph with the sentences

"*However, the estimates of the individual models varied considerably. Considering all three gases and the largest countries, the scatter was smallest for the UK (1σ standard deviation of 3-11%), followed by France (8-15%), Germany (19-22%), Italy (12-31%), and Spain+Portugal (24-30%). The individual models often did not overlap within the range of the combined uncertainties suggesting that ..*"

and in the last paragraph with

"*The network has the potential to identify significant shortcomings in the nationally reported emissions but a denser network would be needed for a more accurate assignment to individual countries. Model-to-model differences were often very large whereas the model median appears to have significant skill as judged from the comparison with reported HFC-134a emissions, which are considered to be relatively well known.*"

Nevertheless, to better emphasize the wide range of country estimates, we will replace the standard uncertainties of the means by the standard deviations of the individual estimates (in percent of the mean) and add another sentence on typical ranges between minimum and maximum. The sentences in the 3rd last paragraph will read as follows:

*Considering all three gases and the largest countries and defining "scatter" by the 1σ standard deviation of individual estimates (in % of the mean), the scatter was smallest for the UK (5-22%), followed by France (16-28%), Germany (38-43%), Italy (23-63%), and Spain+Portugal (42-51%). Differences between minimum and maximum estimates for a given country were often as large as a factor 2, sometimes even a factor of 3, especially for Italy and Spain+Portugal.*

Furthermore, the last sentence in the conclusions will be changed to

*Model-to-model differences were often very large, **occasionally as large as the estimated emissions**, whereas the median appears to ..*

It is difficult to provide a useful statistics summarizing the results of an ensemble of only 4 models. Nevertheless, the median is more robust than the mean value and is commonly used for model ensembles. Note that we also show the full range of the model estimates (in Fig. 13), not only the medians.

**A detail that came to my attention is that the release height for the particles at Jungfraujoch was adjusted for the NAME model to match the FLEXPART footprints. Essentially, this adjustment appears arbitrary and contradicts the general philosophy to use production settings for each model. If the adjustment was not made (transport induced) model differences would be even larger. So, given that (at least one of) the transport models are not able to correctly model transport at the mountain sites, how confident are you with respect to your overall conclusions?**

Unlike for FLEXPART, we did not do any independent analysis on the best release height for the NAME model. Previous analysis provided an optimum release height for FLEXPART. Instead, we used a release height for NAME that produced model time series as close as possible to FLEXPART's given a specific emissions field. We will explain this in the text. This approach allowed us to include the results of NAME despite of the difficulties in representing this mountain site. A thorough investigation of the reasons for the differences between FLEXPART and NAME for Jungfraujoch would be desirable, but was not feasible within the scope of this project.

**P2,L24: regulated reported -> reported**

Done

**P9,L6 and following: Occasionally, I got confused by the naming conventions. I would suggest using NAME and FLEXPART when referring to transport issues and the others names when referring to the entire modelling systems: P9,L6: UKMO -> NAME, P9,L13: NAME->UKMO, check other places.**

We changed the sentence that confused the reviewer to:

*In particular, the score of the NAME-based system UKMO is moving closer to the three FLEXPART-based systems EMPA, EMPA2, and NILU.*

---

## Author Comment (AC3) · 14 Jun 2017

**Reply to referee#3**

The authors would like to thank anonymous referee #3 for the careful review and the helpful comments.

In the following, the reviewer's comments will be in **bold** font, and the responses will be in plain font, with suggested new text in *italics*.

**The authors used four inverse models to estimate European emissions of HFC-134a, HFC-125 and SF6 for the year 2011. All systems used measurements from Jungfraujoch, Mace Head, and Monte Cimone. The paper is well written and provides interesting insights. I think the main problem of the paper was that the differences in the choices, such as spatial correlations of the prior and background treatment, had a quite substantial impact on differences among the models. What was the reason that those were not controlled? If they were better controlled, maybe we could have had more insights on which models are doing better and what we might do to improve the emissions estimation through inverse modeling. Below are some other comments and questions I had and I would recommend publication after they are addressed.**

The main motivation was to document the uncertainty associated with the different choices that have been made in recent halocarbon inversion studies. There is no doubt that differences would have been substantially smaller with a more strongly controlled setup. It was not our intention to assess the quality of the transport simulations of FLEXPART as compared to NAME, which it would ultimately come down to if all other choices were identical.

**For Figure 1, is this the sensitivity created using FLEXPART or NAME? I would also assume that the sensitivity is quite different depending on the month. Which month is this? And is this the monthly mean?**

We apologize that the figure caption was not sufficiently clear (as also noted by another reviewer). It will be changed to

*Annual mean surface sensitivity in units of [ppb/(kg m-2 s-1)] for (a) the original 0.1°x0.1°grid and (b) for the reduced grid of the FLEXPART-based model system EMPA.*

**It was a little unclear why NAME needed such a high release height at Jungfraujoch. If the point of the paper is to better understand the differences among the four inversion systems, I find it puzzling that the authors would modify to make the model footprint sensitivities comparable to each other.**

As also mentioned in our reply to reviewer #2, the measurements from Jungfraujoch have not been used in previous inversion studies based on the NAME model, because the results had not been satisfactory for this site and no independent analysis on the optimal release height had been conducted before, in contrast to FLEXPART. The approach chosen here was pragmatic, so as to not disadvantage the NAME model, and allowed us to include the results of NAME despite of these difficulties. A thorough investigation of the reasons for the differences between FLEXPART and NAME for Jungfraujoch would be desirable, but was not feasible within the scope of this project.

**I had a hard time understanding how the emissions were created following the country outlines. What was the means used to split the EDGAR grid to country outlines? Also, because the prior emissions are so different, I find it more informative if the Fig. 6 was not comparing between prior and posterior but EDGAR and posterior.**

We will add the information that the original resolution of EDGAR was 0.1° x 0.1°. In the case of the UKMO system, EDGAR emissions were first regridded to a fine grid, and the country outlines were then followed as closely as possible. Each grid cell was assigned to the country with the largest share. Except for small countries, the error introduced by this procedure with cells at the borders shared by

more than one country is small. We extended the sentence explaining the grid for the UKMO system with

*follows the country outlines* **as closely as possible given the resolution of a fine grid underlying the reduced inversion grid**.

EDGAR is the prior. Note that the prior emissions are not different, only their spatial representation. Comparing the results with EDGAR at the original resolution would require redistributing the emissions estimated on the reduced grid to the original fine grid. This would be doable technically, but it would give a wrong impression of high resolution of the inverted emissions. We strongly prefer the present representation of the results.

**Why did EMPA2 use the uncertainty set uniformly to 137%? This seemed a little strange and was curious for the reason behind this specific value.**

The 137% was a result of the requirement, that the total uncertainty of a domain covering most of Europe was 20%. We will add this information to the text.

**One of the explanations for why UK's estimated emissions are much higher than what is reported to UNFCCC, the authors mention the use of an assumed high loss rate of HFC-134a from car air conditioning systems in the UK. Why is this only in the UK and how different is the loss rate among the countries? Is a similar explanation possible for overestimation and/or underestimation for different species?**

The UK inventory is conservative (overestimates), as it assumes that there is a 100% replacement of air conditioning fluid in all mobile air conditioning systems each year. Each country makes their own choice in this aspect provided it is backed by expert knowledge. It is not clear what every country across Europe does in this respect. This particular situation is specific for HFC-134a but other issues will undoubtedly impact the emissions of different countries for different gases.

**Backwards mode time differ substantially among the models and I would have expected UKMO to have larger difference between prior and posterior away from the measurement sites, compared to the other model systems that have shorter time span. Why is it that UKMO shows almost no difference between the two farther away from the measurement sites?**

A backward simulation over 5 days captures most of the sensitivities of the measurements to emissions within Europe because the sensitivity decreases very rapidly as time and distance from the measurement increases. Extending to 10 days (NILU) or even 19 days (UKMO) changes little. The fact that UKMO adjusts relatively little at larger distances from the sites must be due to the specific choices of a priori versus observation uncertainties.

**Minor comments**

**1. Sometimes authors state the country by name and sometimes by the ISO2 convention country code. It is a little confusing to me and so I would suggest to be consistent and I would appreciate if there was a table listing the country names with ISO2 code if the authors want to use the codes.**

Instead of adding another table we added the country names in the caption of Figure 11:

*CH=Switzerland, DE=Germany, IT=Italy, FR=France, ES=Spain, PT=Portugal, UK=United Kingdom, IR=Ireland, BE=Belgium, NL=Netherlands, LU=Luxemburg, AT=Austria, DK=Denmark, SW=Sweden, FI=Finland, PO=Poland, CZ=Czech Republic, SV=Slovakia, NO=Norway.*

**2. P. 11 l. 4 "An important question is the context … is the question" -> delete the second "the question" in the sentence to make it "… Paris Agreement is, how suitable is…"**

Thank you, done

**3. I am not quite sure what 0.1°x0.1°min means in Table 1.**

It means that the minimum size of a grid cell is 0.1°x0.1°, but larger when cells are aggregated for the reduced grid. We will change "min." to "minimum".

**4. "reduced acc. to" -> "reduced according to" in Table 1 for UKMO**

Done

**5. State vector length is mentioned in Table 1 but was not explained in the text at all. Can this be clarified in terms of how this is used in the equation and why the equations look so different depending on the system?**

We don't quite agree: The state vector $x$ was introduced in the context of equation 1 as follows: "*$x$ is the state vector which includes the gridded emissions and possibly other elements such as background mole fractions, and n is the number of state vector elements to be estimated/optimized by the inversion*." The state vector was also referred to at other locations, e.g. in the sentence "*In the EMPA system, a single element per observation site is added to the state vector to represent the background at time step k.*"

To better link the information in Table 1 with the text we will add the following line after the first sentence mentioned above:

*An overview of the number and type of state vector elements used in each system is provided in Table 1.*

Note that the equations presented in the manuscript do not depend on the specific choices of state vector elements.

**6. How is the EDGAR prior uncertainty determined in Figure 13? I find that to be a little misleading, since I do not think EDGAR provides such a value.**

EDGAR indeed does not report uncertainties. These uncertainties denote the range of uncertainties in the prior that is introduced by the different gridding methods. This information will be added to the caption.

**7. Figure 14 is very difficult to see – maybe a different color scheme would work better.**

We played a lot with different color schemes and found that using a single color in different shadings works best. The main issue is that the different grid shapes already introduce a lot of variation, such that the figures become too complex and less easily readable when using a color scheme composed of multiple colors.